# Revisiting Data Augmentation in Deep Reinforcement Learning

**Jianshu Hu, Yunpeng Jiang**
UM-SJTU Joint Institute
Shanghai Jiao Tong University
Shanghai, China
`{hjs1998,jyp9961}@sjtu.edu.cn`

**Paul Weng**
Data Science Research Center
Duke Kunshan University
Kunshan, Jiangsu, China
`paul.weng@duke.edu`

## Abstract

Various data augmentation techniques have been recently proposed in image-based deep reinforcement learning (DRL). Although they empirically demonstrate the effectiveness of data augmentation for improving sample efficiency or generalization, which technique should be preferred is not always clear. To tackle this question, we analyze existing methods to better understand them and to uncover how they are connected. Notably, by expressing the variance of the Q-targets and that of the empirical actor/critic losses of these methods, we can analyze the effects of their different components and compare them. We furthermore formulate an explanation about how these methods may be affected by choosing different data augmentation transformations in calculating the target Q-values. This analysis suggests recommendations on how to exploit data augmentation in a more principled way. In addition, we include a regularization term called tangent prop, previously proposed in computer vision, but whose adaptation to DRL is novel to the best of our knowledge. We evaluate our proposition[1] and validate our analysis in several domains. Compared to different relevant baselines, we demonstrate that it achieves state-of-the-art performance in most environments and shows higher sample efficiency and better generalization ability in some complex environments.

## 1 Introduction

Although Deep Reinforcement Learning (DRL) has shown its effectiveness in various tasks, like playing video games (Mnih et al., 2013) and solving control tasks (Li et al., 2019), it still suffers from low sample efficiency and poor generalization ability. To tackle those two issues, data augmentation, a proven simple and efficient technique in computer vision (Kumar et al., 2023), starts to be actively studied in image-based DRL where it has been implemented in various ways (Laskin et al., 2020; Kostrikov et al., 2020; Raileanu et al., 2021). However, most such work has been mainly experimental and a more principled comparison between these state-of-the-art methods is lacking.

In this paper, we study data augmentation techniques in image-based online DRL. We formulate a general actor-critic scheme integrating data augmentation. We then show that current data augmentation methods, categorized as explicit/implicit regularization, are instances of this general scheme. In explicit regularization, image transformations are used in regularization terms calculated with augmented samples to explicitly enforce invariance in the actor and critic. By contrast, image transformations are directly applied on the observations during training in implicit regularization.

Following the analysis about implicit and explicit regularization, we propose a principled data augmentation method in DRL and further justify its design. We start the justification with a discussion about applying different image transformations in calculating the target Q-values. Hansen et al. (2021) propose to avoid using complex image transformations in calculating the targets to stabilize the training. We provide further analysis on why some image transformations are complex and not suitable for calculating the target and how to judge whether an image transformation is complex or not.

---

[1]The source code of our method: `https://github.com/Jianshu-Hu/drqv2`

In addition, we justify other components of our method (e.g., KL regularization in policy) by analyzing the variance of the actor/critic losses and the variance of the Q-estimation under image transformation, which is important to control since applying random image transformations necessarily increase the variance of those statistics. The analysis also reveals the importance of learning the invariance in critic for stabilizing the training. This latter observation motivates us to include an adaption of tangent prop (Simard et al., 1991) regularization in the training of the critic.

**Contributions:** (1) We analyze existing state-of-the-art data augmentation methods in image-based DRL and show how they are related. (2) We provide an empirical and theoretical analysis for the different components in these methods to justify their effectiveness. (3) Based on our analysis, we propose a principled data augmentation actor-critic scheme, which also includes tangent prop, which is novel in DRL. (4) We empirically validate our analysis and evaluate our method.

## 2    RELATED WORK

Many data augmentation techniques for DRL have been proposed, mostly for image-based DRL (Ma et al., 2022), although the fully-observable setting has also been considered Lin et al. (2020). Data augmentation can be used to generate artificial observations, transitions, or trajectories for an existing DRL algorithm or for improving representation learning. Most propositions investigate the usual online DRL training, but recently data augmentation with self-supervised learning in DRL (Srinivas et al., 2020; Schwarzer et al., 2021) has become more active following its success in computer vision (He et al., 2020; Grill et al., 2020). For space reasons, we focus our discussion on the most relevant methods for our work: data augmentation for image-based online DRL.

The first methods to leverage data augmentation in DRL apply transformations directly on observations to generate artificial ones to train the RL agent, which can lead to better generalization (Cobbe et al., 2018). In particular, Reinforcement learning with Augmented Data (RAD) (Laskin et al., 2020) extends Soft Actor-Critic (SAC) (Haarnoja et al., 2018) and Proximal Policy Optimization (PPO) (Schulman et al., 2017) to directly train with augmented observations. Instead, Data-regularized Q (DrQ) (Kostrikov et al., 2020) introduces in SAC the idea of using an averaged Q-target by leveraging more augmented samples. Using Deep Deterministic Policy Gradient (DDPG) (Lillicrap et al., 2015), DrQ-v2 (Yarats et al., 2021) provides various implementation and hyperparameter optimizations and gives up the idea of averaged Q-target. SVEA (Hansen et al., 2021) avoids using complex image transformation when calculating the target values because doing so increases the variance of the target value. Raileanu et al. (2021) argue that simply applying image transformations on the observations in PPO may lead to a wrong estimation of the actor loss and instead propose adding regularization terms in the actor and critic losses. They also propose some methods of automatically choosing image transformation. In recent work like (Liu et al., 2023; Yuan et al., 2022), they investigate problems of using a trainable image transformation which is orthogonal to ours.

## 3    BACKGROUND

In this section, we define notations, recall invariant transformations in DRL, formulate a generic actor-critic scheme exploiting data augmentation, and explain how existing methods fit this scheme according to how data augmentation is applied in the actor and critic losses.

**Notations**    For any set $\mathcal{X}$, $\Delta(\mathcal{X})$ denotes the set of probability distributions over $\mathcal{X}$. For a random variable $X$, $\mathbb{E}[X]$ (resp. $\mathbb{V}[X]$) denotes its expectation (resp. variance). For any function $\phi$ and i.i.d. samples $x_1, \dots, x_N$ of $X$, $\hat{\mathbb{E}}[\phi(X)] = \frac{1}{N} \sum_{i=1}^{N} \phi(x_i)$ is an empirical mean estimating $\mathbb{E}[\phi(X)]$.

A Markov Decision Process (MDP) $M = (\mathcal{S}, \mathcal{A}, r, T, \rho_0)$ is composed of a set of state $\mathcal{S}$, a set of action $\mathcal{A}$, a reward function $r : \mathcal{S} \times \mathcal{A} \to \mathbb{R}$, a transition function $T : \mathcal{S} \times \mathcal{A} \to \Delta(\mathcal{S})$, and a probability distribution over initial states $\rho_0 \in \Delta(\mathcal{S})$. In reinforcement learning (RL), the agent is trained by interacting with the environment to learn a policy $\pi(\cdot \mid s) \in \Delta(\mathcal{A})$ such that the expected return $\mathbb{E}_\pi[\sum_{t=0}^{\infty} \gamma^t r_t \mid s_0 \sim \rho_0]$ is maximized.

---

**Algorithm 1** Data-Augmented Off-policy Actor-Critic Scheme

---

**Hyperparameters**: total number of training steps $T$, mini-batch size $N$, policy update frequency $\kappa$, image transformation set $\mathcal{F}_{\mathcal{T}} = \{f_\tau \mid \tau \in \mathcal{T}\}$, distributions $\mathrm{P}, \mathrm{B} \in \Delta(\mathcal{T})$, random variables $\nu \sim \mathrm{P}, \mu \sim \mathrm{B}$ to transform states.

1: Initialize critic $Q_\phi(s, a)$, actor $\pi_\theta(s)$, and empty replay buffer $\mathcal{D}$.
2: Start with initial state $s_0$.
3: **for** $t = 0 \ldots T$ **do**
4:     Interact with the environment using action from current policy $a_t \sim \pi(\cdot \mid s_t)$.
5:     Save transition $(s_t, a_t, r_t, s_{t+1})$ in replay buffer $\mathcal{D}$.
6:     **if** episode ends **then** reset $s_{t+1}$ to an initial state **end if**
7:     // Actor-critic update
8:     Sample mini-batch $\{(s_i, a_i, r_i, s'_i) \mid i = 1, \ldots, N\}$ from $\mathcal{D}$.
9:     $L_\phi = \frac{1}{N} \sum_i \ell_\phi(s_i, a_i, r_i, s'_i, \nu, \mu)$
10:    Update $\phi$ (critic parameters) with gradient of $L_\phi$.
11:    **if** $t \bmod \kappa = 0$ **then**
12:       $L_\theta = \frac{1}{N} \sum_i \ell_\theta(s_i, \mu)$
13:       Update $\theta$ (actor parameters) with gradient of $L_\theta$.
14:    **end if**
15: **end for**

---

In image-based RL, the underlying model is actually a partially observable MDP (POMDP): the agent observes images instead of states. However, following common practice, we approximate this POMDP with an MDP by assuming that a state is a stack of several consecutive images. With data augmentation, the same transformation is applied on all the stacked images of that state.

**Invariant Transformations** Data augmentation in image-based control tasks assumes that a set $\mathcal{F}_{\mathcal{T}} = \{f_\tau : \mathbb{R}^{h \times w} \to \mathbb{R}^{h \times w} \mid \tau \in \mathcal{T}\}$ of parameterized ($h \times w$-sized) image transformations $f_\tau$ that leave optimal policies invariant is given for some parameter set $\mathcal{T}$. Exploiting this property in online RL is difficult, since an optimal policy is unknown. However, it suggests to directly focus learning on policies that are invariant with respect to this set $\mathcal{F}_{\mathcal{T}}$. Recall:

**Definition 1** ($\pi$-invariance). *A policy $\pi$ is invariant with respect to an image transformation $f_\tau$ if:*

$$\pi(a \mid s) = \pi(a \mid f_\tau(s)) \text{ for all } s \in \mathcal{S}, a \in \mathcal{A}. \tag{1}$$

Interestingly, the Q-functions of those policies satisfy the following invariance property:

**Definition 2** (Q-invariance). *A Q-function is invariant with respect to image transformation $f_\tau$ if:*

$$Q(s, a) = Q(f_\tau(s), a) \text{ for all } s \in \mathcal{S}, a \in \mathcal{A}. \tag{2}$$

The definitions imply that $\mathcal{F}_{\mathcal{T}}$ can be assumed to be closed under composition without loss of generality. We assume that set $\mathcal{T}$ contains $\tau_0$ such that $f_{\tau_0}$ is the identity function, which would allow the possibility of not performing any image transformation. Examples of image transformation are for instance: (small) *random shift*, which pads and then randomly crops the images; *random overlay*, which combines original images with extra images. Empirically, random shift has been shown to be one of the best transformations to accelerate DRL training across a diversity of RL tasks, while random overlay is helpful for generalization.

We formulate a simple actor-critic scheme (Algorithm 1), which boils down to standard off-policy actor-critic if the original losses are applied. Existing data-augmentation-based DRL methods fit this scheme by enforcing Q-invariance and/or $\pi$-invariance via **explicit** or **implicit** regularization, as explained next. We discuss them next and explain how they fit Algorithm 1. Due to the page limit, we recall all the related DRL algorithms in Appendix A and consider SAC (Haarnoja et al., 2018) as the base algorithm in the main text and discuss the variant with DDPG (Lillicrap et al., 2015) in the corresponding appendices.

### 3.1 EXPLICIT REGULARIZATION

The invariant transformations $\mathcal{F}_{\mathcal{T}}$ can be directly used to promote the invariance of the learned Q-function and learn more invariant policies with respect to them. Formally, this can be achieved

by formulating the following (empirical) critic loss and actor loss with two explicit regularization terms: for a transition $(s, a, r, s')$ and random variables $\nu, \mu$ over $\mathcal{T}$,

$$\ell_\phi^E(s, a, r, s', \nu) = \Big(Q_\phi(s, a) - y(s', a')\Big)^2 + \alpha_Q \hat{\mathbb{E}}_\nu[(Q_\phi(f_\nu(s), a) - Q_{\phi, sg}(s, a))^2] \text{ and} \quad (3)$$

$$\ell_\theta^E(s, \mu) = \alpha \log \pi_\theta(\hat{a} \mid s) - Q_\phi(s, \hat{a}) + \alpha_\pi \hat{\mathbb{E}}_\mu\Big[D_{KL}\big(\pi_{\theta, sg}(\cdot \mid s) \,\|\, \pi_\theta(\cdot \mid f_\mu(s))\big)\Big], \quad (4)$$

where $\phi$, $\theta$ are the parameter of the critic $Q_\phi$ and actor $\pi_\theta$, target $y(s', a')$ is defined as $r + \gamma Q_{\bar{\phi}}(s', a') - \alpha \log \pi_\theta(a'|s')$ with $a' \sim \pi_\theta(\cdot \mid s')$ and target network parameter $\bar{\phi}$, action $\hat{a}$ is sampled from $\pi_\theta(\cdot \mid s)$, coefficients $\alpha, \alpha_Q$, and $\alpha_\pi$ respectively correspond to the entropy term and the regularization terms to promote invariance in the critic and actor, and subscript $sg$ represents "stop gradient" for the corresponding term.

In practice, Equations 3 and 4 could be used in an actor-critic algorithm: for instance, they could replace the losses in lines 9 and 12 in Algorithm 1. Note that DrAC (Raileanu et al., 2021) uses them by setting the distribution of $\nu$ and $\mu$ to be uniform (although DrAC is based on the PPO algorithm).

## 3.2 IMPLICIT REGULARIZATION

Another commonly used data augmentation method in DRL is directly applying the image transformation on states during training. Formally, for a transition $(s, a, r, s')$ and random variables $\nu, \mu$ over $\mathcal{T}$, the (empirical) critic and actor losses with implicit regularization are respectively:

$$\ell_\phi^I(s, a, r, s', \nu, \mu) = \hat{\mathbb{E}}_\nu\Big[\Big(Q_\phi(f_\nu(s), a) - \hat{\mathbb{E}}_\mu[y(f_\mu(s'), a')]\Big)^2\Big] \text{ and} \quad (5)$$

$$\ell_\theta^I(s, \mu) = \hat{\mathbb{E}}_\mu\Big[\alpha \log \pi_\theta(\hat{a} \mid f_\mu(s)) - Q_\phi(f_\mu(s), \hat{a})\Big], \quad (6)$$

with $y(f_\mu(s'), a') = r + \gamma Q_{\bar{\phi}}(f_\mu(s'), a') - \alpha \log \pi_\theta(a'|s'), a' \sim \pi_\theta(\cdot \mid f_\mu(s'))$, and $\hat{a} \sim \pi_\theta(\cdot \mid f_\mu(s))$.

The empirical means can be calculated with different numbers of samples: $M$ and $K$ samples for $\hat{\mathbb{E}}_\nu$ and $\hat{\mathbb{E}}_\mu$ in the critic loss, and $J$ samples for $\hat{\mathbb{E}}_\mu$ in the actor loss. Note that with $K > 1$, the corresponding empirical expectation corresponds to the average target used in DrQ. If $\nu$ and $\mu$ have a uniform distribution, Equations 5 and 6 are the losses used in DrQ (Kostrikov et al., 2020) with $J = 1$ and RAD (Laskin et al., 2020) with $M = 1, K = 1$, and $J = 1$. Moreover, different image transformations for $\nu$ and $\mu$ can be used. In SVEA (Hansen et al., 2021), complex image transformations are only applied on states $s$ and included in random variable $\nu$ in the critic loss.

## 4 THEORETICAL DISCUSSION

In this section, we theoretically compare explicit regularization with implicit regularization.

## 4.1 CRITIC LOSS

Before comparing the critic losses in these two regularizations, we first compare an alternative explicit regularization one may think of to enforce Q-invariance, which uses:

$$\ell_\phi^E(s, a, r, s', \nu) = \Big(Q_\phi(s, a) - y(s', a')\Big)^2 + \alpha_Q \hat{\mathbb{E}}_\nu\Big[\Big(Q_\phi(f_\nu(s), a) - y(s', a')\Big)^2\Big]. \quad (7)$$

The only difference compared with Equation 3 is to replace the target in the regularization term with $y(s', a')$. Intuitively, using $y(s', a')$ is a better target because it not only promotes the invariance in the critic, but also serves as a target to improve the critic. In Appendix C.1, we show that using $y(s', a')$ leads to a smaller bias which may be preferred in explicit regularization.

Now the critic losses in explicit regularization and implicit regularization can be connected by setting the distributions of the image transformation parameters, as shown in the following simple lemma:

**Lemma 1.** (C.2)[2] *There exist distributions for $\hat{\nu}$ and $\hat{\mu}$ such that we have for any sample $(s, a, r, s')$:*

$$(\alpha_Q + 1)\ell_\phi^I(s, a, r, s', \hat{\nu}, \hat{\mu}) = \ell_\phi^E(s, a, r, s', \nu)$$

Note that the extra factor $\alpha_Q + 1$ can be controlled by adjusting the learning rate of gradient descent. Therefore, explicit regularization amounts to assigning a higher probability to $\tau_0$.

---

[2]All detailed derivations/proofs are in the appendix. The appendix number is provided for ease of reference.

## 4.2 ACTOR LOSS

Obviously, using the actor loss in explicit regularization (Equation 4) promotes the invariance in the policy. More interestingly, the actor loss in implicit regularization (Equation 6) also implicitly enforces policy invariance, but under some conditions, as we show next. First, recall that like in SAC, the target policy used in the actor loss for state $s$ can be defined as $g(a \mid s) = \exp(\frac{1}{\alpha} Q_\phi(s, a) - \log Z(s))$, in which $Z$ is the partition function for normalizing the distribution. We can prove that the actor loss in implicit regularization can be rewritten with a KL regularization term:

**Proposition 4.1.** *(B.1) Assume that the critic is invariant with respect to transformations in $\mathcal{F}_\mathcal{T}$, i.e., $Q_\phi(f_\tau(s), a) = Q(s, a)$ for all $\tau \in \mathcal{T}, a \in \mathcal{A}$. For any random variable $\mu$, the actor loss in implicit regularization (Equation 6) can be rewritten:*

$$\ell_\theta^I(s, \mu) = \hat{\mathbb{E}}_\mu \Big[ \int_a \pi_\theta(a \mid f_\mu(s)) \log \frac{\pi_{\theta, sg}(a \mid s)}{g(a \mid s)} \Big] + \hat{\mathbb{E}}_\mu \Big[ D_{KL} \Big( \pi_\theta(\cdot \mid f_\mu(s)) \, \| \, \pi_{\theta, sg}(\cdot \mid s) \Big) \Big]. \quad (8)$$

Intuitively, with the actor loss in implicit regularization, the policy $\pi_\theta(\cdot \mid f_\mu(s))$ for each augmented state is trained to get close to its target policy $g(\cdot \mid f_\mu(s))$, which is completely defined by the critic. When the invariance of the critic has been learned, the $\pi_\theta(\cdot \mid f_\mu(s))$'s for different augmented states are actually trained to get close to the same target policy, which then induces policy invariance.

Interestingly, the direction of the KL divergences in Equation 4 and Equation 8 is reversed with respect to the stop gradient. The KL divergence in Equation 4 should be preferred because the other one also implicitly maximizes the policy entropy (see Appendix D), which leads to less stable training, as we have observed empirically. Since policy invariance is only really enforced when the critic is already invariant and that the implicitly enforced KL divergence has opposite direction, it seems preferable to directly enforce policy invariance like in explicit regularization (Equation 4).

Following the idea of using averaged target in the critic loss, another interesting question about applying a KL regularization is whether to use a policy $\pi_{\theta, sg}(\cdot \mid f_\eta(s))$ of a transformed state as the target:

$$\ell_\theta^E(s, \mu) = \alpha \log \pi_\theta(\hat{a} \mid s) - Q_\phi(s, \hat{a}) + \alpha_\pi \hat{\mathbb{E}}_\mu \Big[ \hat{\mathbb{E}}_\eta \big[ D_{KL}(\pi_{\theta, sg}(\cdot \mid f_\eta(s)) \, \| \, \pi_\theta(\cdot \mid f_\mu(s))) \big] \Big], \quad (9)$$

where $\eta$ and $\mu$ follow the same distribution. Intuitively, the regularization term in this equation enforces the invariance across policies of different transformed states and somehow is equivalent to using an averaged policy as the target. As shown in Appendix C.3, the loss in Equation 9 is an upper bound of the loss with an averaged policy $\pi_{avg}(\cdot \mid s) = \hat{\mathbb{E}}_\eta[\pi_{\theta, sg}(\cdot \mid f_\eta(s))]$ as the target:

$$\ell_\theta^E(s, \mu) = \alpha \log \pi_\theta(\hat{a} \mid s) - Q_\phi(s, \hat{a}) + \alpha_\pi \hat{\mathbb{E}}_\mu \Big[ D_{KL}(\pi_{avg}(\cdot \mid s) \, \| \, \pi_\theta(\cdot \mid f_\mu(s))) \Big]. \quad (10)$$

So using Equation 9 is somehow equivalent to using an averaged policy as the target in the KL regularization term and thus has an effect of smoothing, which may be preferred.

## 5 PRINCIPLED DATA-AUGMENTED OFF-POLICY ACTOR-CRITIC ALGORITHM

Following these analyses, we finally propose our generic algorithm and provide a justification for it.

### 5.1 GENERIC ALGORITHM

Formally, our generic algorithm follows the structure in Algorithm 1 but with the following (empirical) critic and actor losses, $\ell_\phi(s, a, r, s', \nu, \mu)$ and $\ell_\theta(s, \mu)$, for a single transition $(s, a, r, s')$ [3]:

$$\ell_\phi = \sum_{i=1}^{n_T} \alpha_i \hat{\mathbb{E}}_{\nu_i} \left[ \Big( Q_\phi(f_{\nu_i}(s), a) - \hat{\mathbb{E}}_\mu[y(f_\mu(s'), a')] \Big)^2 \right] + \alpha_{tp} \hat{\mathbb{E}}_\mu[\nabla_\mu Q_\phi(f_\mu(s), a)] \text{ and} \quad (11)$$

$$\ell_\theta = \hat{\mathbb{E}}_\mu \Big[ \alpha \log \pi_\theta(\hat{a} \mid f_\mu(s)) - Q_\phi(f_\mu(s), \hat{a}) + \alpha_\pi \hat{\mathbb{E}}_\eta \big[ D_{KL}(\pi_{\theta, sg}(\cdot \mid f_\eta(s)) \, \| \, \pi_\theta(\cdot \mid f_\mu(s))) \big] \Big], \quad (12)$$

---

[3] To simplify notations, we drop the parameters of the losses when there is no risk of confusion.

where $n_T$ is the number of types of image transformations used in the training, $f_{\nu_i}$ (resp. $f_\mu, f_\eta$) corresponds to the transformation parameterized by random variable $\nu_i$ (resp. $\mu, \eta$) and $\alpha_i$ (resp. $\alpha_{tp}, \alpha_\pi$) is the coefficient of the mean-squared errors for different image transformations (resp. tangent prop term, KL regularization term). Note that $\eta$ and $\mu$ follow the same distribution.

## 5.2 JUSTIFICATION FOR THE GENERIC ALGORITHM

In this section, we justify further the formulation of our generic algorithm by analyzing first the effects of applying different image transformations in calculating the target Q-values and then the estimation of the empirical actor/critic losses under data augmentation. While the variances of these estimations are unavoidably increased when random image transformations are applied, they can be controlled by several techniques, as we discuss below. Recall that a lower variance of an empirical loss implies a lower variance of its corresponding stochastic gradient, which leads to less noisy updates and may therefore stabilize training. Moreover, a lower variance of target Q-values entails a lower variance of the empirical critic loss, which then similarly leads to more stable training.

**Applying image transformations in calculating the target** A simple technique to reduce the variance of an estimator is to use more samples. Recall that in contrast to RAD, DrQ leverages more augmented samples and introduces the average target, which can be interpreted as using more augmented samples for estimating the target Q-values. In Appendix E.1, we show that DrQ enjoys a smaller variance of the empirical critic loss and the target Q-values than RAD, which may explain the empirical better performance of DrQ (see e.g., (Kostrikov et al., 2020)).

However, when facing complex image transformations, Hansen et al. (2021) propose to avoid using them in calculating the target Q-values to reduce the variance of the target and thus stabilize the training. Interestingly, we observe experimentally that even using complex image transformation such as random conv in the target does not induce a large variance in the target while a much larger bias is observed for the trained critic, as shown in Appendix F. Compared to using random shift which is a finite set of image transformations, using random conv actually applies an infinite set of transformations. To maintain the invariance among augmented states, more updates are required to train the agent sufficiently. To validate the idea that the invariance with respect to complex image transformations is harder to learn and more updates are required when using them in calculating the target, we evaluate increasing the number of updates in each iteration when using different image transformations in calculating the target, as shown in Appendix F. From the experimental results, we observe that the cosine similarity between the augmented features at early stage (e.g., 100k training steps) could be used as a criterion for judging if an image transformation is complex or not. Moreover, for those complex image transformations, updating more is helpful when we apply them in calculating the target Q-values.

**Adding KL regularization** The analysis in Section 4.2 suggests that explicitly adding a KL regularization term, $D_{\eta,\mu} = D_{KL}(\pi_\theta(\cdot \,|\, f_\eta(s)) \,\|\, \pi_\theta(\cdot \,|\, f_\mu(s)))$ with $\eta$ and $\mu$ following the same distribution over $\mathcal{F}_\mathcal{T}$, in the actor loss can help better learn the invariance of the actor. Thus, we define the actor loss in the generic algorithm as the sum of the actor loss in implicit regularization (Equation 6) and this KL regularization term. Below, we provide two additional justifications for this choice.

Firstly, we show in the following proposition that the variance of the actor loss in implicit regularization can be controlled by a KL divergence if the invariance is already learned for critic $Q_\phi$. Hence, adding a KL regularization term in the actor loss may both enforce invariance and reduce the variance of the implicit actor loss.

**Proposition 5.1.** *(E.2) Assuming that critic $Q_\phi$ is invariant with respect to transformations in $\mathcal{F}_\mathcal{T}$, we have:*

$$\mathbb{V}_\mu[\ell_\theta^I(s,\mu)] \leq \frac{1}{n}\mathbb{E}_\mu\left[\left(\mathbb{E}_\eta[D_{\eta,\mu} + c(f_\nu(s))\sqrt{2D_{\eta,\mu}}]\right)^2\right], \tag{13}$$

*where $c(f_\nu(s)) > 0$ and $n$ is the number of samples to estimate the empirical mean $\ell_\theta^I(s,\mu)$.*

In addition, under invariance of the target critic $Q_{\bar\phi}$, we can prove that the variance of target Q-values can be controlled by a KL divergence. Therefore, training with a KL regularization term may further reduce the variance of the target, which would also consequently reduce the variance of the critic loss.

**Proposition 5.2.** *(E.3) Assuming that target critic $Q_{\bar{\phi}}$ is invariant with respect to transformations in $\mathcal{F}_{\mathcal{T}}$, we have*

$$\mathbb{V}_{\mu}[\hat{Y}(s',\mu)] \leq \frac{1}{n}\mathbb{E}_{\mu}\left[\left(\mathbb{E}_{\eta}\left[\max_{a'}(y(f_{\mu}(s'),a')-r)\sqrt{2D_{\eta,\mu}}+\alpha \cdot D_{\eta,\mu}\right]\right)^2\right] \quad (14)$$

*where target $\hat{Y}(s',\mu) = \hat{\mathbb{E}}_{\mu}[\mathbb{E}_{a'\sim\pi_{\theta}(\cdot|f_{\mu}(s'))}[y(f_{\mu}(s'),a')]]$, $y(f_{\mu}(s'),a') = r + \gamma Q_{\bar{\phi}}(f_{\mu}(s'),a') - \alpha\log\pi(a'|f_{\mu}(s'))$ and $n$ is the number of samples to estimate the empirical mean $\hat{Y}(s',\mu)$.*

Although those results are obtained under invariance of the (current or target) critics, we may expect an approximate reduction of the variances by adding a KL regularization term, when the critics are approximately invariant. Experimentally, we can observe that the variance of target and critic losses can be reduced thanks to this KL regularization term.

**Tangent Prop**   We now introduce an additional regularization to promote the invariance of the critic. This addition is motivated by our previous analysis, which indicates that the critic invariance is an important factor for stabilizing training. This additional regularization term is based on tangent prop (Simard et al., 1991), which was proposed for computer vision tasks, to promote invariance by enforcing that the gradient of the trained model with respect to the magnitude of image transformation be zero. In DRL, when applied to the critic, it can be formulated as follows.

If the invariance of Q-function over the whole set of the image transformation parameter $\tau$ is required and the Q-function is differentiable with respect to $\tau$, tangent prop regularization is actually adding the constraints on the derivative of the Q-function with respect to $\tau$:

$$\frac{\partial Q(f_{\tau}(s),a)}{\partial \tau} = \frac{\partial Q(f_{\tau}(s),a)}{\partial f_{\tau}(s)}\frac{\partial f_{\tau}(s)}{\partial \tau} \approx \frac{\partial Q(f_{\tau}(s),a)}{\partial f_{\tau}(s)}\frac{f_{\tau+\delta\tau}(s)-f_{\tau}(s)}{\delta\tau} = 0. \quad (15)$$

Although the computational cost is increased due to the differentiation of $Q$, this regularization is still beneficial since sample efficiency is often one of main concerns when applying DRL. Compared to the original tangent prop, we also extend it to a broader application by applying it not only on the original state but also on the transformed states. Specifically, instead of only calculating the derivative around $\tau_0$, the derivative is estimated at any $\tau \in \mathcal{T}$. A theoretical justification for how tangent prop regularization is related to the training of the critic can be found in Appendix B.2.

## 6   EXPERIMENTAL RESULTS

In order to validate our theoretical analysis and show the effectiveness of our proposed algorithm, we perform a series of experiments to (1) experimentally validate our propositions, (2) conduct a case study explicitly showing the statistics we analyzed, (3) compare our final proposed algorithm with state-of-the-art baselines (RAD, DrAC, DrQ, DrQv2, SVEA) to verify its sample efficiency, and evaluate its generalization ability against SVEA, which was specifically-designed for this purpose.

**Experimental Set Up**   We evaluate different methods on environments from **DeepMind Control Suite** (Tassa et al., 2018) with normal background for evaluating sample efficiency and distracted background for evaluating generalization ability. **DeepMind Control Suite** is a commonly used benchmark for evaluating methods of applying data augmentation in DRL. Across different environments, all hyperparameters are listed in Appendix G such as learning rates and batch size for the actor and critic. All experiments are performed with 5 different random seeds and the agent is evaluated every 10k environment steps, whose performance is measured by cumulative rewards averaged over 10 evaluation episodes.

**Ablation study**   To validate our propositions, we first compare RAD [M=1,K=1], RAD+ [M=2,K=1] and DrQ [M=2,K=2] using the official implementation from DrQ [4], where $[M, K]$ are respectively the numbers of samples used for estimating the empirical mean over random variables $[\nu, \mu]$ in the critic loss. We then show the effectiveness of KL regularization and tangent prop regularization by adding them one by one based on DrQ. All these methods are trained with normal

---
[4]`https://github.com/denisyarats/drq`

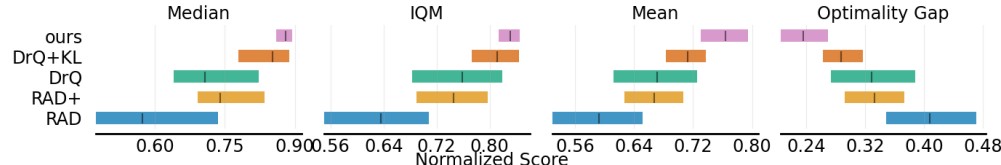

Figure 1: Aggregated results of validating our propositions.

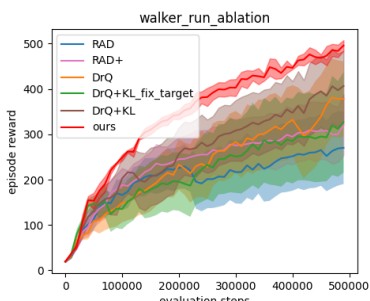

Figure 2: Performance of different methods in walker run environment.

| Methods | | Std of critic loss | | Std of target Q | | Std of actor loss | | KL |
|---|---|---|---|---|---|---|---|---|
| RAD | | 0.772 ±0.359 | | 0.321 ±0.084 | | 4.971 ±1.044 | | 0.266 ±0.015 |
| RAD+ | (a) | 0.271 ±0.043 | | 0.234 ±0.031 | | 5.011 ±0.463 | | 0.248 ±0.021 |
| DrQ | | 0.358 ±0.123 | (a) | 0.225 ±0.055 | | 5.077 ±0.964 | | 0.293 ±0.055 |
| DrQ+KL(fix target) | | 0.384 ±0.042 | | 0.221 ±0.007 | | 5.708 ±1.259 | | 0.135 ±0.009 |
| DrQ+KL | | 0.319 ±0.053 | (b) | 0.183 ±0.018 | (b) | 4.469 ±0.745 | (b) | 0.099 ±0.010 |
| ours | (c) | 0.280 ±0.004 | (c) | 0.182 ±0.001 | | 4.815 ±0.412 | | 0.101 ±0.007 |

Table 1: Important statistics of different methods in walker run environment. (a)(b)(c) are discussed in the main text.

background using random shift as the image transformation and also evaluated with normal background. When applicable, we adopt hyperparameters from (Kostrikov et al., 2020) except that we reduce the batch size from 512 to 256 (to make the algorithms easier to run on our computing device, equipped with one NVIDIA RTX 3060 GPU and Intel i7-10700 CPU). Considering that only one image transformation (random shift) is applied, the weight $\alpha_i$ for it is set as 1 in Equation 11. We tune the weights $\alpha_{KL}$ and $\alpha_{tp}$ based on a quick grid search in $\{0.1, 0.5, 1.0\}$ and finally choose 0.1 for both $\alpha_{KL}$ and $\alpha_{tp}$. The results of validating our proposition are shown in Figure 1. We follow the recommendations from Agarwal et al. (2021) to plot the aggregated performance over totally 9 environments. Detailed training curves in each environments are included in Appendix H.1.

**Case study** We conduct a comprehensive evaluation in walker-run using the same setting as above to answer the following questions: 1) Does our proposition help reduce the variance we are concerned about? 2) Does our proposition help learn the invariance in the feature space? Moreover, we run the experiment of using a fixed target in the KL regularization to validate our analysis in Section 4.2.

Note that the set of parameters $\mathcal{T}$ is finite when using random shift as image transformation. So, augmented Q-values $Q_\phi(f_\tau(s), a)$ and augmented target Q-values $Q_{\bar{\phi}}(f_{\tau'}(s'), a')$ for all $\tau, \tau' \in \mathcal{T}$ are recorded within a mini-batch for every 10k environment steps. These are used to calculate the standard deviation of the empirical critic loss and the target Q-values. The KL divergence between policies for two augmented samples is also recorded with the same frequency. The results for this case study are shown in Figure 2 and Table 1. From the table, we can see (a) the variance of the critic loss and the variance of the target Q decrease thanks to more augmented samples, (b) KL regularization helps enforce the invariance in the actor and reduce the variance of the target and the variance of the actor loss, (c) tangent prop further improves learning the invariance in the critic.

Meanwhile, the feature vectors of augmented samples within a mini-batch after the encoder from the actor and critic are recorded with the same frequency. To estimate the invariance in the feature space, we first calculate the cosine similarities between the augmented features. We also apply t-SNE (van der Maaten & Hinton, 2008) to project the latent features into 2D space and calculate the $L_2$-distance between the projected points. With these measures, the learned invariance of the actor and critic features along the training are shown in Appendix H.2. Following our proposition, the agent can learn well the invariance in the feature spaces of both the actor and critic. From another view, this achieves the same goal as using the self-supervised learning losses (Srinivas et al., 2020; Grill et al., 2020) to explicitly enforce the invariance in the latent space.

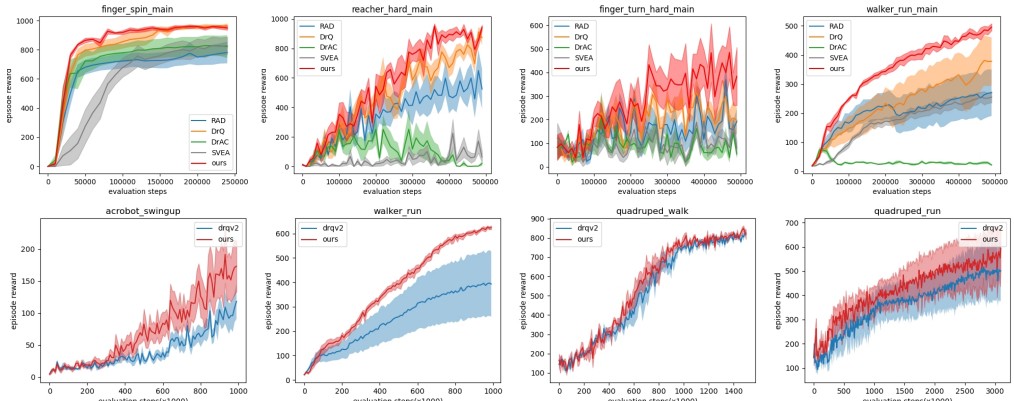

Figure 3: Partial results of evaluating sample efficiency for our methods.

| Environments | SVEA | ours |
|---|---|---|
| walker walk | 449.52±60.33 | **508.28±79.82** |
| walker stand | 788.80±89.31 | **843.27±51.33** |
| cartpole swingup | 351.96±62.36 | **366.77±58.08** |
| ball in cup catch | 445.04±145.42 | **569.50±142.99** |
| finger spin | 338.35±77.65 | **447.99±73.52** |

Table 2: Comparison of SVEA and ours in DMControl with video-hard backgrounds for training 500k env steps.

| Methods | Std of critic loss | Std of target Q | KL |
|---|---|---|---|
| SVEA | 5.780 ±1.029 | 0.936 ±0.097 | 0.404 ±0.019 |
| ours | **3.197 ±0.760** | **0.530 ±0.038** | **0.205 ±0.034** |

Table 3: Statistics for the trained model of SVEA and our method using random overlay in walker walk environment.

**Main result** We evaluate our proposed methods in sample efficiency and generalization ability. For sample efficiency, we evaluate our method (ours) against state-of-the-art baselines: RAD, DrQ, DrAC (with SAC as the base algorithm), DrQv2 (with DDPG as the base algorithm) and SVEA (with random overlay). The experimental results shown in Figure 3 confirm that our method outperforms previous ones. Due to the page limit, the additional evaluations are included in Appendix H.3.

For generalization, we compare SVEA with our method based on the official implementation of SVEA[5]. Considering that random overlay shows the best generalization ability among the image transformations listed in SVEA, our comparison also includes random overlay. When applicable, we adopt the same hyperparameter values as in SVEA. Thus, we use the same $\alpha_i = 0.5$ for random shift and random overlay, reuse $\alpha_{KL} = 0.1$ from previous experiments and tune $\alpha_{tp}$ based on a quick grid search in $\{0.1, 0.5\}$ and finally choose 0.5 for $\alpha_{tp}$. Both methods (SVEA and ours) are trained in normal background for 500k environment steps, and evaluated in video-hard backgrounds. The final results comparing SVEA and our method are shown in Table 2. We can see the improvement in generalization especially in environments such as ball in cup catch, finger spin, and walker walk. Meanwhile, similar statistics we discussed before are recorded, as shown in Table 3. The complete curves for evaluations and the recorded statistics are listed in Appendices H.4 and H.5.

## 7 CONCLUSION

We revisit state-of-the-art data augmentation methods in DRL. Our theoretical analysis helps understand and compare different existing methods. In addition, this analysis provides recommendations on how to exploit data augmentation in a more theoretically-motivated way. We included tangent prop, a regularization term to promote invariance, which is novel in DRL. We validated our propositions and evaluated our method in DeepMind control tasks with normal background for comparing sample efficiency and distracted background for comparing generalization ability with the state-of-the-art methods. Limitations are discussed in Appendix I due to the page limit.

---

[5] https://github.com/nicklashansen/dmcontrol-generalization-benchmark

**Reproducibility** For the experimental results, the code with comments on how to reproduce the results is released, the experimental settings are described in the main paper (Section 6) and the hyperparameters required for reproducing the results are recorded in Appendix G. For the theoretical results, all the detailed proofs can be found in the appendix.

**Acknowledgments** This work has been supported in part by the program of National Natural Science Foundation of China (No. 62176154).

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

## A  BACKGROUNDS

**Deep Deterministic Policy Gradient**  Incorporating a parameterized actor function $\mu_\theta(s)$, Deep Deterministic Policy Gradient uses the following actor and critic and loss to train the agent:

$$
\begin{aligned}
J_\pi(\theta) &= \hat{\mathbb{E}}_{s_t \sim \mathcal{D}}[-Q_\phi(s_t, a_t)|_{a_t=\mu_\theta(s_t)}], \\
J_Q(\phi) &= \hat{\mathbb{E}}_{s_t \sim \mathcal{D}}[(Q_\phi(s_t, a_t) - \hat{Q}(s_t, a_t))^2|_{a_t=\mu_\theta(s_t)}],
\end{aligned}
\tag{16}
$$

where $\hat{Q}(s_t, a_t) = r_t + \gamma Q_{\bar{\phi}}(s_{t+1}, \mu_{\bar{\theta}}(s_{t+1}))$, which is the target Q-value defined from a target network, $\theta$ and $\phi$ represents the parameters of the actor and the critic respectively, $\bar{\theta}$ and $\bar{\phi}$ represents the parameters of the target actor and the target critic respectively, and $\mathcal{D}$ represents the replay buffer. The weights of a target network are the exponentially moving average of the online network's weights.

**Soft Actor-Critic**  Maximum entropy RL tackles an RL problem with an alternative objective function, which favors more random policies: $J = \hat{\mathbb{E}}_\pi[\sum_{t=0}^\infty \gamma^t r_t + \alpha H(\pi(\cdot \mid s_t))]$, where $\gamma$ is the discount factor, $\alpha$ is a trainable coefficient of the entropy term and $H(\pi(\cdot \mid s_t))$ is the entropy of action distribution $\pi(\cdot \mid s_t)$. The Soft Actor-Critic (SAC) algorithm (Haarnoja et al., 2018) optimizes it by training the actor $\pi_\theta$ and critic $Q_\phi$ with the following respective losses:

$$
\begin{aligned}
J_\pi(\theta) &= \hat{\mathbb{E}}_{s_t \sim \mathcal{D}, a \sim \pi}[\alpha \log \pi_\theta(a \mid s_t) - Q_\phi(s_t, a)], \\
J_Q(\phi) &= \hat{\mathbb{E}}_{s_t, a_t \sim \mathcal{D}}[(Q_\phi(s_t, a_t) - \hat{Q}(s_t, a_t))^2],
\end{aligned}
\tag{17}
$$

where $\hat{Q}(s_t, a_t) = r_t + \gamma Q_{\bar{\phi}}(s_{t+1}, a_{t+1}) - \alpha \log \pi_\theta(a_{t+1}|s_{t+1})$, which is the target Q-value defined from a target network and $a_{t+1} \sim \pi_\theta(\cdot \mid s_{t+1})$, $\theta$, $\phi$ and $\bar{\phi}$ represents the parameters of the actor, the critic and the target critic respectively, and $\mathcal{D}$ represents the replay buffer. The weights of a target network are the exponentially moving average of the online network's weights.

**Reinforcement Learning with Augmented Data**  Reinforcement Learning with Augmented Data (RAD) (Laskin et al., 2020) applies data augmentation in SAC by replacing the original observation with augmented observations in the training of the actor and critic. Given image transformation $f_\nu$, the actor and critic losses are

$$
\begin{aligned}
J_\pi(\theta) &= \hat{\mathbb{E}}_{s_t \sim \mathcal{D}, a \sim \pi}[\alpha \log \pi_\theta(a \mid f_\nu(s_t)) - Q_\phi(f_\nu(s_t), a)], \\
J_Q(\phi) &= \hat{\mathbb{E}}_{s_t, a_t \sim \mathcal{D}}[(Q_\phi(f_\nu(s_t), a_t) - \hat{Q}(f_\nu(s_t), a_t))^2],
\end{aligned}
\tag{18}
$$

**Data-Regularized Q**  Data-regularized Q (DrQ) (Kostrikov et al., 2020) extends RAD by using data augmentation in the training of the critic in two new ways. Given a type of image transformation $f$ parameterized by $\nu$, data augmentation is first applied in the calculation of the target Q-value for every transition $(s, a, r, s')$:

$$
y = r + \gamma \frac{1}{K} \sum_{k=1}^K Q_{\bar{\phi}}(f_{\tau_k}(s'), a'), \text{ where } a' \sim \pi(\cdot \mid f_{\tau_k}(s')).
\tag{19}
$$

$Q_{\bar{\phi}}$ is the slowly updated target network. Then the critic is updated with different augmented $s$ and this averaged target:

$$
\ell_Q(\phi) = \frac{1}{NM} \sum_{i=1}^N \sum_{m=1}^M \left(Q_\phi(f_{\tau_m}(s), a) - y\right)^2.
\tag{20}
$$

Note that DrQ recovers RAD when $M = 1$ and $K = 1$.

**SVEA**  In order to avoid non-deterministic Q-target and over-regularization, Hansen et al. (2021) propose using state without complex augmentation for calculating the target. Let $\mathcal{T}_1$ and $\mathcal{T}_2$ be a set of random shift and a set of random shift plus one of the data augmentation mentioned in the paper such as random convolution (Lee et al., 2019). The critic loss used for training is

$$
L_Q(\phi) = \frac{1}{N} \sum_{i=1}^N \alpha_{\text{svea}}(Q_\phi(f_{\tau_{1,i}}(s), a) - y_i)^2 + \beta_{\text{svea}}(Q_\phi(f_{\tau_{2,i}}(s), a) - y_i)^2,
\tag{21}
$$

where $\alpha_{\text{svea}}$ and $\beta_{\text{svea}}$ are constant coefficients for naively and complexly augmented data respectively, $\tau_{1,i} \in \mathcal{T}_1$ and $\tau_{2,i} \in \mathcal{T}_2$, $y_i = r + \gamma Q_{\bar{\phi}}(\tau_{1,i}(s'), a') + \alpha \log \pi(a' \mid f_{\tau_{1,i}}(s'))$, where $a' \sim \pi(\cdot \mid \tau_{1,i}(s'))$.

**DrAC** Instead of directly replacing the original samples with augmented samples in the training, Raileanu et al. (2021) use two regularization terms in the training of the actor and critic to explicitly enforce the invariance. When applying it in the PPO algorithm (Schulman et al., 2017) to learn a state-value estimator $V_\phi(s)$ and a policy $\pi_\theta(s)$, the regularization terms are

$$
\begin{aligned}
G_V &= (\hat{V}(s) - V_\phi(f_\nu(s)))^2, \\
G_\pi &= D_{KL}[\pi_\theta(a \mid s) \mid \pi_\theta(a \mid f_\nu(s)].
\end{aligned}
\tag{22}
$$

where $\hat{V}(s)$ is the sum of rewards collected by the agent after state $s$ and $\nu$ is the random variable for parameterizing the image transformation.

**Tangent Prop Regularization** Tangent prop (Simard et al., 1991) is a regularization term used for learning invariance for a function $G(s)$ with respect to a small image transformation parameterized by $\alpha$ on $s$:

$$
\sum \left\| \frac{\partial G(s, \alpha)}{\partial \alpha} \right\|^2 = 0
\tag{23}
$$

# B  EXPECTED LOSS UNDER DATA AUGMENTATION

## B.1  ACTOR LOSS

**SAC as base algorithm** For image-based control tasks, a data augmentation $f$ parameterized by $\mu$ over $\mathcal{T}$ is applied on the observations. The actor loss with implicit regularization for the state $s$ in a transition is

$$
\begin{aligned}
\ell_\theta^I(s, \mu) &= \hat{\mathbb{E}}_\mu \Big[ \alpha \log \pi_\theta(\hat{a} \mid f_\mu(s)) - Q_\phi(f_\mu(s), \hat{a}) \mid_{\hat{a} \sim \pi_\theta(\cdot \mid f_\mu(s))} \Big] \\
&= \hat{\mathbb{E}}_\mu \Big[ D_{KL} \Big( \pi_\theta(\cdot | f_\mu(s)) || \exp(\frac{1}{\alpha} Q_\phi(f_\mu(s), \cdot) - \log Z(f_\mu(s))) \Big) \Big]
\end{aligned}
\tag{24}
$$

Let $g(f_\mu(s), \cdot) = \exp(\frac{1}{\alpha} Q_\phi(f_\mu(s), \cdot) - \log Z(f_\mu(s)))$.

$$
\begin{aligned}
&\ell_\theta^I(s, \mu) \\
&= \hat{\mathbb{E}}_\mu \Big[ D_{KL} \Big( \pi_\theta(\cdot | f_\mu(s)) || g(f_\mu(s), \cdot) \Big) \Big] \\
&= \hat{\mathbb{E}}_\mu \Big[ D_{KL} \Big( \pi_\theta(\cdot | f_\mu(s)) || g(f_\mu(s), \cdot)) \Big) - D_{KL} \Big( \pi_\theta(\cdot | f_\mu(s)) || g(s, \cdot) \Big) \\
&\quad + D_{KL} \Big( \pi_\theta(\cdot | f_\mu(s)) || g(s, \cdot) \Big) \Big] \\
&= \hat{\mathbb{E}}_\mu \Big[ \int_a \pi_\theta(a | f_\mu(s)) \log \frac{\pi_\theta(a | f_\mu(s))}{g(a | f_\mu(s))} - \int_a \pi_\theta(a | f_\mu(s)) \log \frac{\pi_\theta(a | f_\mu(s))}{g(a | s)} \Big] \\
&\quad + \hat{\mathbb{E}}_\mu \Big[ D_{KL} \Big( \pi_\theta(\cdot | f_\mu(s)) || g(s, \cdot) \Big) \Big] \\
&= \hat{\mathbb{E}}_\mu \Big[ \int_a \pi_\theta(a | f_\mu(s)) \log \frac{g(a | s)}{g(a | f_\mu(s))} \Big] + \hat{\mathbb{E}}_\mu \Big[ D_{KL} \Big( \pi_\theta(\cdot | f_\mu(s)) || g(s, \cdot) \Big) \Big] \\
&= \hat{\mathbb{E}}_\mu \Big[ \int_a \pi_\theta(a | f_\mu(s)) \log \frac{g(a | s)}{g(a | f_\mu(s))} \Big] + \hat{\mathbb{E}}_\mu \Big[ D_{KL} \Big( \pi_\theta(\cdot | f_\mu(s)) || g(s, \cdot) \Big) \\
&\quad - D_{KL} \Big( \pi_\theta(\cdot | f_\mu(s)) || \pi_{\theta, sg}(\cdot | s) \Big) + D_{KL} \Big( \pi_\theta(\cdot | f_\mu(s)) || \pi_{\theta, sg}(\cdot | s) \Big) \Big] \\
&= \hat{\mathbb{E}}_\mu \Big[ \int_a \pi_\theta(a | f_\mu(s)) \log \frac{g(a | s)}{g(a | f_\mu(s))} \Big] + \hat{\mathbb{E}}_\mu \Big[ \int_a \pi_\theta(a | f_\mu(s)) \log \frac{\pi_{\theta, sg}(a | s)}{g(a | s)} \Big] \\
&\quad + \hat{\mathbb{E}}_\mu \Big[ D_{KL} \Big( \pi_\theta(\cdot | f_\mu(s)) || \pi_{\theta, sg}(\cdot | s) \Big) \Big].
\end{aligned}
\tag{25}
$$

If the invariance in the critic has been learned:

$$Q(f_\mu(s), a) = Q(s, a) \text{ for all } a \in \mathcal{A}, \tag{26}$$

the actor loss with implicit regularization becomes

$$\ell_\theta^I(s, \mu) = \hat{\mathbb{E}}_\mu \left[ \int_a \pi_\theta(a|f_\mu(s)) \log \frac{\pi_{\theta,sg}(a|s)}{g(a|s)} \right] + \hat{\mathbb{E}}_\mu \left[ D_{KL}\Big( \pi_\theta(\cdot|f_\mu(s)) || \pi_{\theta,sg}(\cdot|s) \Big) \right], \tag{27}$$

because

$$g(a|f_\mu(s)) = g(a|s) \text{ for all } a \in \mathcal{A}. \tag{28}$$

If the actor is well learned for state $s$, this actor loss become

$$\ell_\theta^I(s, \mu) = \hat{\mathbb{E}}_\mu \left[ D_{KL}\Big( \pi_\theta(\cdot|f_\mu(s)) || \pi_{\theta,sg}(\cdot|s) \Big) \right], \tag{29}$$

because

$$g(a|s) = \pi_{\theta,sg}(a|s) \text{ for all } a \in \mathcal{A}. \tag{30}$$

**DDPG as base algorithm** For image-based control tasks, a data augmentation $f$ parameterized by $\mu$ over $\mathcal{T}$ is applied on the observations. Considering that the Q-invariant transformation is also $\pi^*$-invariant, training all policies of the transformed states to get close to the same optimal policy is equivalent to training the policy of original state and enforce the invariance in the policy. Considering actor loss with implicit regularization, we can apply a Taylor expansion with respect to the optimal action $\pi^*(f_\mu(s)) = \pi^*(s) = \arg\max_a Q_\phi(s, a)$:

$$
\begin{aligned}
&\ell_\theta^I(s, \mu) \\
&= \hat{\mathbb{E}}_\mu \Big[ - Q_\phi(f_\mu(s), \hat{a}) \,|_{\hat{a}=\pi_\theta(f_\mu(s))} \Big] \\
&= -\hat{\mathbb{E}}_\mu \Big[ Q_\phi(f_\mu(s), \pi^*(f_\mu(s)) + J(\hat{a} - \pi^*(f_\mu(s))) \\
&\quad + \frac{1}{2}(\hat{a} - \pi^*(f_\mu(s)))^T H(\hat{a} - \pi^*(f_\mu(s))) + o(\|\hat{a} - \pi^*(f_\mu(s))\|^2)|_{\hat{a}=\pi_\theta(f_\mu(s))} \Big] \\
&\approx -\frac{1}{2}\hat{\mathbb{E}}_\mu \Big[ (\hat{a} - \pi^*(f_\mu(s)))^T H(\hat{a} - \pi^*(f_\mu(s)))|_{\hat{a}=\pi_\theta(f_\mu(s))} \Big] \\
&= -\frac{1}{2}\hat{\mathbb{E}}_\mu \Big[ (\hat{a} - \pi_{\theta,sg}(s) + \pi_{\theta,sg}(s) - \pi^*(f_\mu(s)))^T H \\
&\quad (\hat{a} - \pi_{\theta,sg}(s) + \pi_{\theta,sg}(s) - \pi^*(f_\mu(s)))|_{\hat{a}=\pi_\theta(f_\mu(s))} \Big] \\
&= -\frac{1}{2}\hat{\mathbb{E}}_\mu \Big[ (\hat{a} - \pi_{\theta,sg}(s))^T H(\hat{a} - \pi_{\theta,sg}(s)) \\
&\quad + (\pi_{\theta,sg}(s) - \pi^*(f_\mu(s)))^T H(\pi_{\theta,sg}(s) - \pi^*(f_\mu(s))) \\
&\quad + 2(\hat{a} - \pi_{\theta,sg}(s))^T H(\pi_{\theta,sg}(s) - \pi^*(f_\mu(s)))|_{\hat{a}=\pi_\theta(f_\mu(s))} \Big].
\end{aligned}
\tag{31}
$$

The first term above is enforcing the invariance of the actor with respect to the transformation.

## B.2 CRITIC LOSS

### B.2.1 LINEAR MODEL

According to the analysis by Balestriero et al. (2022), the expected Mean Squared Error (MSE) under data augmentation for a linear regression model can be expressed by the expectation and variance of the transformed images. Now we want to derive a similar regularization term from the critic loss.

If we use linear model for the critic and actor:

$$Q(s, a) = \mathcal{W}_s * s + \mathcal{W}_a * a + b_0 \tag{32}$$

$$\bar{Q}(s, a) = \bar{\mathcal{W}}_s * s + \bar{\mathcal{W}}_a * a + \bar{b}_0 \tag{33}$$

$$\pi(s) = W_\tau * s + \epsilon W_\sigma * s + b_1, \epsilon \sim \mathcal{N}(0, 1) \tag{34}$$

in which $\mathcal{W}_s \in \mathcal{R}^{1*|\mathcal{S}|}$, $\mathcal{W}_a \in \mathcal{R}^{1*|\mathcal{A}|}$, $\mathcal{W}_\tau \in \mathcal{R}^{1*|\mathcal{S}|}$, $\mathcal{W}_\sigma \in \mathcal{R}^{1*|\mathcal{S}|}$ and $b_0$, $b_1$ are parameters for the model. $\bar{Q}$ is the exponential moving average of $Q$.

The critic loss for a transition $(s, a, r, s')$ under data augmentation $\nu \sim \mathcal{P}$ and $\mu \sim \mathcal{P}'$ for state $s$ and next state $s'$ is

$$
\begin{aligned}
\ell_\phi &= \mathbb{E}_\nu\left[\left(Q(f_\nu(s), a) - \hat{\mathbb{E}}_\mu[y]\right)^2\right] \\
&= \mathbb{E}_\nu\left[Q(f_\nu(s), a)^2\right] - 2\mathbb{E}_\nu\left[Q(f_\nu(s), a)\right]\hat{\mathbb{E}}_\mu[y] + \hat{\mathbb{E}}_\mu[y]^2)
\end{aligned}
\tag{35}
$$

in which

$$y = r + \gamma\bar{Q}(f_\mu(s'), a') - \alpha\log\pi(a'|f_\mu(s'))|_{a'\sim\pi(\cdot|f_\mu(s'))}. \tag{36}$$

Considering the last term is not used to update $Q$, we only need to focus on the first two terms.

**Expectation**

$$
\begin{aligned}
\mathbb{E}_\nu\left[Q(f_\nu(s), a)\right] &= \mathbb{E}_\nu[\mathcal{W}_s f_\nu(s) + \mathcal{W}_a a + b_0] \\
&= \mathcal{W}_s\mathbb{E}_\nu[f_\nu(s)] + \mathcal{W}_a a + b_0 \\
&= Q(\mathbb{E}_\nu[f_\nu(s)], a)
\end{aligned}
\tag{37}
$$

**Variance**

$$
\begin{aligned}
&\mathbb{E}_\nu\left[Q(f_\nu(s), a)^2\right] \\
&= \mathbb{E}_\nu\left[(\mathcal{W}_s f_\nu(s) + \mathcal{W}_a a + b_0)^2\right] \\
&= \mathbb{E}_\nu\left[f^T(s, \nu)\mathcal{W}_s^T\mathcal{W}_s f_\nu(s) + (\mathcal{W}_a a + b_0)^2 + 2(\mathcal{W}_a a + b_0)(\mathcal{W}_s f_\nu(s))\right] \\
&= \mathbb{E}_\nu\left[Tr\left(\mathcal{W}_s^T\mathcal{W}_s f_\nu(s)f^T(s, \nu)\right)\right] + (\mathcal{W}_a a + b_0)^2 + 2(\mathcal{W}_a a + b_0)(\mathcal{W}_s\mathbb{E}_\nu[f_\nu(s)]) \\
&= Tr\left(\mathcal{W}_s^T\mathcal{W}_s\mathbb{E}_\nu\left[f_\nu(s)f^T(s, \nu)\right]\right) + (\mathcal{W}_a a + b_0)^2 + 2(\mathcal{W}_a a + b_0)(\mathcal{W}_s\mathbb{E}_\nu[f_\nu(s)]) \\
&= Tr\left(\mathcal{W}_s^T\mathcal{W}_s\left(\mathbb{E}_\nu\left[f_\nu(s)f^T(s, \nu)\right] - \mathbb{E}_\nu\left[f_\nu(s)\right]\mathbb{E}_\nu\left[f^T(s, \nu)\right]\right)\right) \\
&\quad + Tr\left(\mathcal{W}_s^T\mathcal{W}_s\mathbb{E}_\nu\left[f_\nu(s)\right]\mathbb{E}_\nu\left[f^T(s, \nu)\right]\right) + (\mathcal{W}_a a + b_0)^2 + 2(\mathcal{W}_a a + b_0)(\mathcal{W}_s\mathbb{E}_\nu[f_\nu(s)]) \\
&= Tr\left(\mathcal{W}_s^T\mathcal{W}_s\mathbb{V}_\nu[f_\nu(s)]\right) + Q(\mathbb{E}_\nu[f_\nu(s)], a)^2
\end{aligned}
\tag{38}
$$

**Whole loss**

$$
\begin{aligned}
\ell_\phi &= \sum_{i=1}^N \mathbb{E}_\nu\left[Q(f_\nu(s), a)^2\right] - 2\mathbb{E}_\nu\left[Q(f_\nu(s), a)\right]\hat{\mathbb{E}}_\mu[y] + \hat{\mathbb{E}}_\mu[y]^2 \\
&= \sum_{i=1}^N Tr\left(\mathcal{W}_s^T\mathcal{W}_s\mathbb{V}_\nu[f_\nu(s)]\right) + Q(\mathbb{E}_\nu[f_\nu(s)], a)^2 - 2\hat{\mathbb{E}}_\mu[y]Q(\mathbb{E}_\nu[T_\nu(s)], a) + \hat{\mathbb{E}}_\mu[y]^2 \\
&= \sum_{i=1}^N \left(Q(\mathbb{E}_\nu[f_\nu(s)], a) - \hat{\mathbb{E}}_\mu[y]\right)^2 + Tr\left(\mathcal{W}_a^T\mathcal{W}_a\mathbb{V}_\nu[f_\nu(s)]\right)
\end{aligned}
\tag{39}
$$

### B.2.2 NON-LINEAR MODEL

According to the analysis by Balestriero et al. (2022), the expected loss of transformed state has an upper bound related to the variance of the transformed state:

$$\mathbb{E}[(\ell \circ Q)(f(x))] \leq (\ell \circ Q)(\mathbb{E}[f(x)]) + \kappa(x)\|\mathcal{J}Q(\mathbb{E}[f(x)])H(x)\Lambda(x)^{\frac{1}{2}}\|_F^2, \tag{40}$$

in which variance of the transformed image can be decomposed into

$$\mathbb{V}[f(x)] = H(x)\Lambda(x)H(x)^T. \tag{41}$$

The second term in the RHS of Equation 40 recovers tangent prop regularization.

## C EXPLICIT VS IMPLICIT REGULARIZATION

### C.1 CRITIC LOSS IN EXPLICIT REGULARIZATION

For $\tau = (s, a, s', r)$ sampled from the replay buffer $\mathcal{D}$, given current estimation $Q_\phi$ and true estimation $Q^*$ without error, the bias of the target $\mathbb{E}_{a' \sim \pi(s')} y(s', a')$ is smaller than the target $Q_{\phi, sg}(s, a)$:

$$
\begin{aligned}
&\mathbb{E}_{\tau \sim \mathcal{D}} \Big[ (\mathbb{E}_{a' \sim \pi(\cdot|s')}[y(s', a')] - Q^*(s, a))^2 \Big] \\
=& \mathbb{E}_{\tau \sim \mathcal{D}} \Big[ \Big( \mathbb{E}_{a' \sim \pi(\cdot|s')}[r + \gamma Q_{\bar{\phi}}(s', a') - \alpha \log \pi(a'|s')] \\
& \quad - (r + \gamma \mathbb{E}_{a' \sim \pi(\cdot|s')}[Q^*(s', a') - \alpha \log \pi(a'|s')] \Big)^2 \Big] \\
=& \mathbb{E}_{\tau \sim \mathcal{D}} \Big[ \Big( \gamma \mathbb{E}_{a' \sim \pi(\cdot|s')}[Q_{\bar{\phi}}(s', a') - Q^*(s', a')] \Big)^2 \Big] \\
\approx& \gamma^2 \mathbb{E}_{\tau \sim \mathcal{D}} \Big[ \Big( \mathbb{E}_{a' \sim \pi(\cdot|s')}[Q_{\phi, sg}(s', a') - Q^*(s', a')] \Big) \Big] \\
<& \mathbb{E}_{\tau \sim \mathcal{D}} \Big[ \Big( \mathbb{E}_{a' \sim \pi(\cdot|s')}[Q_{\phi, sg}(s', a') - Q^*(s', a')] \Big)^2 \Big] \\
<& \mathbb{E}_{\tau \sim \mathcal{D}, a' \sim \pi(\cdot|s')} \Big[ \Big( Q_{\phi, sg}(s', a') - Q^*(s', a') \Big)^2 \Big] \\
=& \mathbb{E}_{\tau \sim \mathcal{D}} \Big[ \Big( Q_{\phi, sg}(s, a) - Q^*(s, a) \Big)^2 \Big]
\end{aligned}
\tag{42}
$$

The bias of using a target $\bar{y}$ in the explicit regularization is

$$
\begin{aligned}
\epsilon(\bar{y}) =& \mathbb{E}_\tau \Big[ \Big( \Big( Q_\phi(f_\nu(s), a) - \bar{y} \Big)^2 - \Big( Q_\phi(f_\nu(s), a) - Q^*(s, a) \Big)^2 \Big)^2 \Big] \\
=& \mathbb{E}_\tau \Big[ \Big( 2 Q_\phi(f_\nu(s), a)(Q^*(s, a) - \bar{y}) + \bar{y}^2 - Q^*(s, a)^2 \Big)^2 \Big].
\end{aligned}
\tag{43}
$$

We only need to consider the first term $2Q_\phi(f_\nu(s), a)(Q^* - \bar{y})$, considering that other terms is constant with respect to $\phi$. So the bias of using different targets in the regularization term is decided by the bias of the target compared to the true estimation. The bias of using $\mathbb{E}_{a' \sim \pi(s')}[y(s', a')]$ in the explicit regularization term is smaller than using $Q_{\phi, sg}(s, a)$ according to the equations above:

$$
\epsilon(\bar{y} = \mathbb{E}_{a' \sim \pi(\cdot|s')}[y(s', a')]) < \epsilon(\bar{y} = Q_{\phi, sg}(s, a)),
\tag{44}
$$

In practice, we use the sampled value $y(s', a')$ as the target, which leads to a smaller bias and relatively larger variance.

### C.2 CRITIC LOSS CONNECTION

Assume given $\ell_\phi^E(s, a, r, s', \nu)$, by appropriately setting the random variables in Equations 5, it recovers the critic loss in explicit regularization (Equation 7), as shown below. If the distributions of $\hat{\nu}$ and $\hat{\mu}$ are defined as follows:

$$
\mathbb{P}(\hat{\nu} = \tau) = \begin{cases} \frac{\mathbb{P}(\nu = \tau)\alpha_Q + 1}{\alpha_Q + 1}, & \text{if } \tau = \tau_0 \\ \frac{\mathbb{P}(\nu = \tau)\alpha_Q}{\alpha_Q + 1}, & \text{if } \tau \neq \tau_0 \end{cases} \quad \mathbb{P}(\hat{\mu}) = \begin{cases} 1, & \text{if } \hat{\mu} = \tau_0 \\ 0, & \text{if } \hat{\mu} \neq \tau_0 \end{cases},
\tag{45}
$$

then we have for any sample $(s, a, r, s')$:

$$
(\alpha_Q + 1)\ell_\phi^I(s, a, r, s', \hat{\nu}, \hat{\mu}) = \ell_\phi^E(s, a, r, s', \nu)
$$

### C.3 ACTOR LOSS

Considering that the policy is parameterized as normal distribution in SAC, we first define:

$$
\pi_{\theta, sg}(\cdot \mid f_\eta(s)) = \mathcal{N}(\lambda_\eta, \sigma_\eta^2), \pi_\theta(\cdot \mid f_\mu(s)) = \mathcal{N}(\lambda_\mu, \sigma_\mu^2)
\tag{46}
$$

For simplicity, we consider $\mu$ and $\eta$ are defined over discrete set $\mathcal{T}$ with probability $P(\mu = \tau_i) = P(\eta = \tau_i) = P_i, \tau_i \in \mathcal{T}$. The derivation can be easily extended to using a continuous set.

$$\pi_{avg}(\cdot \mid s) = \hat{\mathbb{E}}_\eta[\pi_{\theta,sg}(\cdot \mid f_\eta(s))] = \mathcal{N}(\lambda_{avg}, \sigma_{avg}^2) = \mathcal{N}(\sum_i P_i \lambda_{\tau_i}, \sum_i P_i^2 \sigma_{\tau_i}^2) \tag{47}$$

$$
\begin{aligned}
&\hat{\mathbb{E}}_\eta\big[D_{KL}(\pi_{\theta,sg}(\cdot \,|\, f_\eta(s)) \,\|\, \pi_\theta(\cdot \,|\, f_\mu(s))\big] \\
&= \hat{\mathbb{E}}_\eta\big[\log \frac{\sigma_\mu}{\sigma_\eta} + \frac{\sigma_\eta^2}{2\sigma_\mu^2} + \frac{(\lambda_\eta - \lambda_\mu)^2}{2\sigma_\mu^2} - \frac{1}{2}\big] \\
&= \log \sigma_\mu - \sum_i P_i \log \sigma_{\tau_i} + \frac{\sum_i P_i \sigma_{\tau_i}^2}{2\sigma_\mu^2} + \frac{\sum_i P_i(\lambda_{\tau_i} - \lambda_\mu)^2}{2\sigma_\mu^2} - \frac{1}{2}
\end{aligned}
\tag{48}
$$

$$
\begin{aligned}
&D_{KL}(\pi_{avg}(\cdot\,|\,s) \,\|\, \pi_\theta(\cdot\,|\,f_\mu(s)) \\
&= \log \frac{\sigma_\mu}{\sigma_{avg}} + \frac{\sigma_{avg}^2}{2\sigma_\mu^2} + \frac{(\lambda_{avg} - \lambda_\mu)^2}{2\sigma_\mu^2} - \frac{1}{2} \\
&= \log \sigma_\mu - \frac{1}{2}\log \sum_i P_i^2 \sigma_{\tau_i}^2 + \frac{\sum_i P_i^2 \sigma_{\tau_i}^2}{2\sigma_\mu^2} + \frac{(\sum_i P_i \lambda_{\tau_i} - \lambda_\mu)^2}{2\sigma_\mu^2} - \frac{1}{2}
\end{aligned}
\tag{49}
$$

Comparing the two equations above, the first term and the last term are the same, and the second term is a constant with respect to the parameter $\theta$ of the actor. For the third term, it is obvious that $\frac{\sum_i P_i \sigma_{\tau_i}^2}{2\sigma_\mu^2} \geq \frac{\sum_i P_i^2 \sigma_{\tau_i}^2}{2\sigma_\mu^2}$ because $P_i \geq P_i^2$, for any $i$. For the forth term, we have:

$$
\begin{aligned}
&\frac{\sum_i P_i(\lambda_{\tau_i} - \lambda_\mu)^2}{2\sigma_\mu^2} - \frac{(\sum_i P_i \lambda_{\tau_i} - \lambda_\mu)^2}{2\sigma_\mu^2} \\
&= \frac{\sum_i P_i(\lambda_{\tau_i}^2 + \lambda_\mu^2 - 2\lambda_{\tau_i}\lambda_\mu)}{2\sigma_\mu^2} - \frac{\lambda_\mu^2 + (\sum_i P_i \lambda_{\tau_i})^2 - 2\lambda_\mu \sum_i P_i \lambda_{\tau_i}}{2\sigma_\mu^2} \\
&= \frac{\lambda_\mu^2 + \sum_i P_i \lambda_{\tau_i}^2 - 2\lambda_\mu \sum_i P_i \lambda_{\tau_i}}{2\sigma_\mu^2} - \frac{\lambda_\mu^2 + (\sum_i P_i \lambda_{\tau_i})^2 - 2\lambda_\mu \sum_i P_i \lambda_{\tau_i}}{2\sigma_\mu^2} \\
&= \frac{\sum_i P_i \lambda_{\tau_i}^2 - (\sum_i P_i \lambda_{\tau_i})^2}{2\sigma_\mu^2} = \frac{\mathbb{V}[\lambda_\eta]}{2\sigma_\mu^2} \geq 0
\end{aligned}
\tag{50}
$$

So the loss of using the policy of a transformed state as the target is an upper bound of using the average policy as the target:

$$\hat{\mathbb{E}}_\eta\big[D_{KL}(\pi_{\theta,sg}(\cdot\,|\,f_\eta(s)) \,\|\, \pi_\theta(\cdot\,|\,f_\mu(s))\big] \geq D_{KL}(\pi_{avg}(\cdot\,|\,s) \,\|\, \pi_\theta(\cdot\,|\,f_\mu(s))) \tag{51}$$

## D  KL DIVERGENCE

Given a transformation $f_\nu(s)$ on state $s$, considering that the KL divergence is not symmetric, we discuss the differences between two kinds of KL regularization here:

$$D_{KL}\big(\pi_{\theta,sg}(s) \| \pi_\theta(f_\nu(s))\big) \text{ and } D_{KL}\big(\pi_\theta(f_\nu(s)) \| \pi_{\theta,sg}(s)\big), \tag{52}$$

**Detach First**

$$
\begin{aligned}
D_{KL}\big(\pi_{\theta,sg}(s) \| \pi_\theta(f_\nu(s))\big) &= \int_a \pi_{\theta,sg}(a|s) \log \frac{\pi_{\theta,sg}(a|s)}{\pi_\theta(a|f_\nu(s))} \\
&= \int_a \Big(\pi_{\theta,sg}(a|s) \log \pi_{\theta,sg}(a|s) - \pi_{\theta,sg}(a|s) \log \pi_\theta(a|f_\nu(s))\Big) \\
&= -H(\pi_{\theta,sg}(s)) - \int_a \pi_{\theta,sg}(a|s) \log \pi_\theta(a|f_\nu(s))
\end{aligned}
\tag{53}
$$

**Detach Second**

$$
\begin{aligned}
D_{KL}\big(\pi_\theta(f_\nu(s))||\pi_{\theta,sg}(s)\big) &= \int_a \pi_\theta(a|f_\nu(s)) \log \frac{\pi_\theta(a|f_\nu(s))}{\pi_{\theta,sg}(a|s)}, \\
&= \int_a \Big(\pi_\theta(a|f_\nu(s)) \log \pi_\theta(a|f_\nu(s)) - \pi_\theta(a|f_\nu(s)) \log \pi_{\theta,sg}(a|s)\Big) \\
&= -H(\pi_\theta(f_\nu(s))) - \int_a \pi_\theta(a|f_\nu(s)) \log \pi_{\theta,sg}(a|s),
\end{aligned}
$$
(54)

in which $H$ represent the entropy for a distribution.

"Detach second" introduces an entropy term for the policy of the transformed state. This regularization not only makes the policy of the augmented state and the original state close, but also maximizes the entropy of the policy of the transformed state. However, in "detach first", the entropy term with the sign $sg$ is not used to update the policy.

# E    VARIANCE UNDER DATA AUGMENTATION

## E.1    MORE AUGMENTED SAMPLES REDUCE THE VARIANCE OF THE CRITIC LOSS

Considering one transition $(s, a, r, s')$ and $M$ transformations $\{f_{\tau_m} \mid m = 1, ..., M\}$, $K$ transformation $\{(f_{\tau'_k} \mid k = 1, ..., K\}$ respectively on $s$ and $s'$, the Q-values and target Q-values for the transformed samples are

$$
\begin{aligned}
Q_m &= Q(f_{\tau_m}(s), a), \\
y_k &= r + \gamma Q_{\bar\phi}(f_{\tau'_k}(s'), a') - \alpha \log \pi(a'|f_{\tau'_k}(s'))|_{a' \sim \pi(\cdot|f_{\tau'_k}(s'))},
\end{aligned}
$$
(55)

where $m \in \{1, ..., M\}, k \in \{1, ..., K\}$.

RAD+ loss becomes

$$
\ell_{RAD+} = \frac{1}{M} \cdot \sum_{m=1}^{M} (Q_m - y_k)^2
$$
(56)

DrQ loss becomes

$$
\begin{aligned}
\ell_{DrQ} &= \frac{1}{M} \cdot \sum_{m=1}^{M} (Q_m - \frac{1}{K}\sum_{k=1}^{K} y_k)^2 \\
&= \frac{1}{M} \cdot \sum_{m=1}^{M} \Big(Q_m^2 + (\frac{1}{K}\sum_{k=1}^{K} y_k)^2 - 2Q_m \cdot \frac{1}{K}\sum_{k=1}^{K} y_k\Big) \\
&= \frac{1}{M} \cdot \sum_{m=1}^{M} Q_m^2 + \frac{1}{M} \cdot \sum_{m=1}^{M} (\frac{1}{K}\sum_{k=1}^{K} y_k)^2 - \frac{1}{M} \cdot \sum_{m=1}^{M} 2Q_m \cdot \frac{1}{K}\sum_{k=1}^{K} y_k \\
&= \frac{1}{M} \cdot \sum_{m=1}^{M} Q_m^2 + \frac{1}{K^2}(\sum_{k=1}^{K} y_j)^2 - \frac{2}{M \cdot K} \cdot \sum_{m=1}^{M}\sum_{k=1}^{K} Q_m \cdot y_k
\end{aligned}
$$
(57)

If all the combinations of above $Q$ and $y$ values are used for estimation, the loss becomes:

$$
\begin{aligned}
\ell_{all} &= \frac{1}{M \cdot K} \cdot \sum_{m=1}^{M}\sum_{k=1}^{K}(Q_m - y_k)^2 \\
&= \frac{1}{M \cdot K} \cdot (\sum_{m=1}^{M} K \cdot Q_m^2 + \sum_{k=1}^{K} M \cdot y_k^2 - 2\sum_{m=1}^{M}\sum_{k=1}^{K}Q_m \cdot y_k) \\
&= \frac{1}{M} \cdot \sum_{m=1}^{M}Q_m^2 + \frac{1}{K}(\sum_{k=1}^{K}y_k)^2 - \frac{2}{M \cdot K} \cdot \sum_{m=1}^{M}\sum_{k=1}^{K}Q_m \cdot y_k
\end{aligned}
\tag{58}
$$

The second terms in both Equation 57 and 58 can be ignored because the gradients of target values $y_j$ with respect to critic parameters are stopped.

Obviously, $\ell_{all}$ and $\ell_{DrQ}$ have same gradients with respect to trainable parameters of the critic. The comparison between $\ell_{DrQ}$ and $\ell_{RAD+}$ is exactly the comparison between $\ell_{all}$ and $\ell_{RAD+}$. For one transition in one gradient step, $M \cdot K$ pairs of $Q_m$ and $y_k$ are used to formulate $\ell_{all}$ while only $M$ pairs of $Q_m$ and $y_k$ are used to formulate $\ell_{RAD+}$. Therefore, we can find out that $DrQ$ outperforms $RAD$ by leveraging more augmented samples and the averaged target. These operations indeed reduce the variance of the estimated critic loss.

### E.2 KL REDUCES THE VARIANCE OF ACTOR LOSS

**SAC actor loss** Given data augmentation $f_\nu | \nu \sim \mathrm{P}$ on state $s$, if $Q(f_\nu(s), a)$ is invariant with respect to $\nu$ for all $a \in \mathcal{A}$, the variance of the actor loss $\mathbb{V}_\nu[\ell_\theta^I(s, \nu)]$ is bounded by a term that depends on the KL divergence $D_{\eta,\nu} = D_{KL}(\pi(\cdot \mid f_\eta(s))\|\pi(\cdot \mid f_\nu(s)))$ for $\nu, \eta \sim \mathrm{P}$:

$$
\mathbb{V}_\nu[\ell_\theta^I(s, \nu)] \leq \frac{1}{n}\mathbb{E}_\nu\left[\left(\mathbb{E}_\eta[D_{\eta,\nu}] + c(f_\nu(s))\sqrt{2D_{\eta,\nu}}\right)^2\right],
\tag{59}
$$

where $c(f_\nu(s)) > 0, n$ is the number of samples to estimate the empirical mean $\ell_\theta^I(s, \nu)$.

*Proof.* For image-based control tasks, a data augmentation $f$ parameterized by $\nu \sim \mathcal{P}$ is applied on the observations. The actor loss of SAC becomes

$$
\ell_\theta^I(s, \nu) = \hat{\mathbb{E}}_\nu\left[D_{KL}\left(\pi_\theta(\cdot|f_\nu(s))\| \exp(\frac{1}{\alpha}Q_\phi(f_\nu(s), \cdot) - \log Z(f_\nu(s)))\right)\right]
\tag{60}
$$

Let $g(f_\nu(s), \cdot) = \exp(\frac{1}{\alpha}Q_\phi(f_\nu(s), \cdot) - \log Z(f_\nu(s)))$.

The variance of empirical mean can be derived as the true variance divided by the number of samples $n$.

$$
\begin{aligned}
\mathbb{V}[\hat{\mathbb{E}}[x]] &= \mathbb{E}[(\hat{\mathbb{E}}(x) - \mathbb{E}[x])^2] \\
&= \mathbb{E}[(\frac{1}{n}(x_1 - \mathbb{E}[x] + x_2 - \mathbb{E}[x] + ... + x_n - \mathbb{E}[x]))^2] \\
&= \frac{1}{n^2}n \cdot \mathbb{V}[x] \\
&= \frac{1}{n}\mathbb{V}[x]
\end{aligned}
\tag{61}
$$

The variance of $\ell_\theta^I(s, \nu)$ with respect to $\nu$ for a given number of samples $n$ is

$$
\begin{aligned}
&\mathbb{V}_\nu[\ell_\theta^I(s, \nu)] \\
&= \frac{1}{n}\mathbb{V}_\nu[D_{KL}(\pi_\theta(\cdot|f_\nu(s))\|g(f_\nu(s), \cdot))] \\
&= \frac{1}{n}\mathbb{E}_\nu\left[\left(D_{KL}(\pi_\theta(\cdot|f_\nu(s))\|g(f_\nu(s), \cdot)) - \mathbb{E}_\eta[D_{KL}(\pi_\theta(\cdot|f_\eta(s))\|g(f_\eta(s), \cdot))]\right)^2\right] \\
&= \frac{1}{n}\mathbb{E}_\nu\left[\left(D_{KL}(\pi_\theta(\cdot|f_\nu(s))\|g(f_\nu(s), \cdot)) - \sum_\eta \mathcal{P}(\eta)D_{KL}(\pi_\theta(\cdot|f_\eta(s))\|g(f_\eta(s), \cdot))\right)^2\right] \\
&= \frac{1}{n}\mathbb{E}_\nu\left[\left(\sum_\eta \mathcal{P}(\eta)(D_{KL}(\pi_\theta(\cdot|f_\eta(s))\|g(f_\eta(s), \cdot)) - D_{KL}(\pi_\theta(\cdot|f_\nu(s))\|g(f_\nu(s), \cdot)))\right)^2\right]
\end{aligned}
\tag{62}
$$

For the term inside the above equation, we can further derive:

$$
\begin{aligned}
&D_{KL}(\pi_\theta(\cdot|f_\eta(s))||g(f_\eta(s),\cdot)) - D_{KL}(\pi_\theta(\cdot|f_\nu(s))||g(f_\nu(s),\cdot)) \\
&= \int_a \pi_\theta(\cdot|f_\eta(s)) \log \frac{\pi_\theta(\cdot|f_\eta(s))}{g(f_\eta(s),\cdot)} - \pi_\theta(\cdot|f_\nu(s)) \log \frac{\pi_\theta(\cdot|f_\nu(s))}{g(f_\nu(s),\cdot)} \\
&= \int_a \pi_\theta(\cdot|f_\eta(s)) \log \pi_\theta(\cdot|f_\eta(s)) - \pi_\theta(\cdot|f_\eta(s)) \log g(f_\eta(s),\cdot) \\
&\quad - \pi_\theta(\cdot|f_\nu(s)) \log \pi_\theta(\cdot|f_\nu(s)) + \pi_\theta(\cdot|f_\nu(s)) \log g(f_\nu(s),\cdot) \\
&= \int_a \pi_\theta(\cdot|f_\eta(s)) \log \frac{\pi_\theta(\cdot|f_\eta(s))}{\pi_\theta(\cdot|f_\nu(s))} - \pi_\theta(\cdot|f_\eta(s)) \log g(f_\eta(s),\cdot) \\
&\quad - (\pi_\theta(\cdot|f_\nu(s)) - \pi_\theta(\cdot|f_\eta(s))) \log \pi_\theta(\cdot|f_\nu(s)) + \pi_\theta(\cdot|f_\nu(s)) \log g(f_\nu(s),\cdot) \\
&= \int_a \pi_\theta(\cdot|f_\eta(s)) \log \frac{\pi_\theta(\cdot|f_\eta(s))}{\pi_\theta(\cdot|f_\nu(s))} - \pi_\theta(\cdot|f_\eta(s)) \log g(f_\nu(s),\cdot) + \pi_\theta(\cdot|f_\eta(s)) \log \frac{g(f_\nu(s),\cdot)}{g(f_\eta(s),\cdot)} \\
&\quad - (\pi_\theta(\cdot|f_\nu(s)) - \pi_\theta(\cdot|f_\eta(s))) \log \pi_\theta(\cdot|f_\nu(s)) + \pi_\theta(\cdot|f_\nu(s)) \log g(f_\nu(s),\cdot) \\
&= D_{KL}(\pi_\theta(\cdot|f_\eta(s))||\pi_\theta(\cdot|f_\nu(s))) \\
&\quad + \int_a \Big(\pi_\theta(\cdot|f_\eta(s)) - \pi_\theta(\cdot|f_\nu(s))\Big) \cdot \Big(\log \pi_\theta(\cdot|f_\nu(s)) - \log g(f_\nu(s),\cdot)\Big) \\
&\quad + \int_a \pi_\theta(\cdot|f_\eta(s)) \log \frac{g(f_\nu(s),\cdot)}{g(f_\eta(s),\cdot)}
\end{aligned}
\tag{63}
$$

Then plug the above results into the equation of $\mathbb{V}_\nu[\ell_\theta^I(s,\nu)]$.

$$
\begin{aligned}
&\mathbb{V}_\nu[\ell_\theta^I(s,\nu)] \\
&= \frac{1}{n}\mathbb{E}_\nu\Big[\Big(\sum_\eta \mathcal{P}(\eta)(D_{KL}(\pi_\theta(\cdot|f_\eta(s))||g(f_\eta(s),\cdot)) - D_{KL}(\pi_\theta(\cdot|f_\nu(s))||g(f_\nu(s),\cdot)))\Big)^2\Big] \\
&= \frac{1}{n}\mathbb{E}_\nu\Big[\Big(\sum_\eta \mathcal{P}(\eta)D_{KL}(\pi_\theta(\cdot|f_\eta(s))||\pi_\theta(\cdot|f_\nu(s))) \\
&\quad + \sum_\eta P(\eta)\int_a (\pi_\theta(\cdot|f_\eta(s)) - \pi_\theta(\cdot|f_\nu(s)))(\log \pi_\theta(\cdot|f_\nu(s)) - \log g(f_\nu(s),\cdot)) \\
&\quad + \sum_\eta P(\eta)\int_a \pi_\theta(\cdot|f_\eta(s)) \log \frac{g(f_\nu(s),\cdot)}{g(f_\eta(s),\cdot)}\Big)^2\Big] \\
&= \frac{1}{n}\mathbb{E}_\nu\Big[\Big(\sum_\eta \mathcal{P}(\eta)D_{KL}(\pi_\theta(\cdot|f_\eta(s))||\pi_\theta(\cdot|f_\nu(s))) \\
&\quad + \sum_\eta P(\eta)\int_a (\pi_\theta(\cdot|f_\eta(s)) - \pi_\theta(\cdot|f_\nu(s))) \cdot (\log \pi_\theta(\cdot|f_\nu(s)) - \log g(f_\nu(s),\cdot)) \\
&\quad + \frac{1}{\alpha}\sum_\eta P(\eta)\int_a \pi_\theta(\cdot|f_\eta(s))(Q_\phi(f_\nu(s),a) - Q_\phi(f_\eta(s),a)) \\
&\quad + \sum_\eta P(\eta)\int_a \pi_\theta(\cdot|f_\eta(s)) \log \frac{Z(f_\nu(s))}{Z(f_\eta(s))}\Big)^2\Big]
\end{aligned}
\tag{64}
$$

For the second term on the right hand side of Equation 64, by applying Pinsker's inequality, we get

$$
\begin{aligned}
\sum_\eta & P(\eta) \int_a (\pi_\theta(\cdot|f_\eta(s)) - \pi_\theta(\cdot|f_\nu(s))) \cdot (\log \pi_\theta(\cdot|f_\nu(s)) - \log g(f_\nu(s), \cdot)) \\
&\leq \sum_\eta \mathcal{P}(\eta) \int_a |\pi_\theta(a|f_\eta(s)) - \pi_\theta(a|f_\nu(s))| \cdot \max_a |\log \pi_\theta(a|f_\nu(s)) - \log g(f_\nu(s), a)| \\
&\leq \sum_\eta \mathcal{P}(\eta) \sqrt{2 D_{KL}\Big(\pi_\theta(\cdot|f_\eta(s)) || \pi_\theta(\cdot|f_\nu(s))\Big)} \cdot \max_a |\log \pi_\theta(a|f_\nu(s)) - \log g(f_\nu(s), a)|
\end{aligned}
\tag{65}
$$

For the third and fourth terms of Equation 64, given data augmentation $f_\nu|\nu \sim \mathrm{P}$ on state $s$, if $Q(f_\nu(s), a)$ is invariant with respect to $\nu$ for all $a \in \mathcal{A}$, both the third and the fourth terms of $\hat{\mathbb{V}}_\nu[\ell_\theta^I(s, \nu)]$ are zero.

Therefore, if $Q(f_\nu(s), a)$ is invariant with respect to $\nu$ for all $a \in \mathcal{A}$, the variance of the augmented actor loss $\mathbb{V}_\nu[\ell_\theta^I(s, \nu)]$ is bounded by the KL divergence $D_{\eta,\nu} = D_{KL}(\pi(\cdot \mid f_\eta(s)) \mid \pi(\cdot \mid f_\nu(s)))$ for $\nu, \eta \sim \mathrm{P}$:

$$
\mathbb{V}_\nu[\ell_\theta^I(s, \nu)] \leq \frac{1}{n} \mathbb{E}_\nu \left[ \left( \mathbb{E}_\eta[D_{\eta,\nu} + c(f_\nu(s)) \sqrt{2 D_{\eta,\nu}}] \right)^2 \right]
\tag{66}
$$

where $c(f_\nu(s)) = \max_a |\log \pi_\theta(a|f_\nu(s)) - \log g(f_\nu(s), a)| > 0$, $n$ is the number of samples to estimate the empirical mean. $\qquad \square$

**DDPG actor loss**    Based on Equation 31, the DDPG actor loss $\ell_\theta^I(s, \mu)$ becomes,

$$
\ell_\theta^I(s, \mu) \approx -\frac{1}{2} \hat{\mathbb{E}}_\mu \left[ (\pi_\theta(f_\mu(s)) - \pi^*(f_\mu(s)))^T H_\mu (\pi_\theta(f_\mu(s)) - \pi^*(f_\mu(s))) \right].
\tag{67}
$$

The variance of the actor loss is reduced if we minimize the mean squared error between two deterministic actions $||\pi_\theta(f_\eta(s')) - \pi_\theta(f_\nu(s'))||^2$, where $\eta, \nu \sim \mathrm{P}$.

*Proof.* Let $M_\mu = \pi_\theta(f_\mu(s)) - \pi^*(f_\mu(s))$. Assuming that the Hessian matrix $H_\mu$ have a a lower bound and upper bound:

$$
l_\mu \mathbf{I} \preceq H_\mu \preceq L_\mu \mathbf{I},
\tag{68}
$$

we have

$$
l_\mu \|M_\mu\|^2 \leq \ell_\theta^I(s, \mu) \leq L_\mu \|M_\mu\|^2.
\tag{69}
$$

$$\mathbb{V}_\nu[\ell_\theta^I(s,\nu)]$$

$$= \frac{1}{n}\mathbb{E}_\nu[(\ell_\theta^I(s,\nu) - \mathbb{E}_\eta[\ell_\theta^I(s,\eta)])^2] = \frac{1}{n}\mathbb{E}_\nu[(\sum_\eta \mathcal{P}(\eta)\ell_\theta^I(s,\nu) - \sum_\eta \mathcal{P}(\eta)\ell_\theta^I(s,\eta))^2]$$

$$= \frac{1}{n}\mathbb{E}_\nu[(\sum_\eta \mathcal{P}(\eta)(\ell_\theta^I(s,\nu) - \ell_\theta^I(s,\eta)))^2] \leq \frac{1}{n}\mathbb{E}_\nu[(\sum_\eta \mathcal{P}(\eta)(\ell_\theta^I(s,\nu) - \ell_\theta^I(s,\eta))^2)]$$

$$= \frac{1}{n}\mathbb{E}_{\nu,\eta}[(\ell_\theta^I(s,\nu) - \ell_\theta^I(s,\eta))^2] \leq \frac{1}{n}\cdot(\max_\nu \ell_\theta^I(s,\nu) - \min_\eta \ell_\theta^I(s,\eta))^2$$

$$\leq \frac{1}{n}\cdot(\max_\nu L_\nu\|M_\nu\|^2 - \min_\eta l_\eta\|M_\eta\|^2)^2 = \frac{1}{n}\cdot(L_{\nu_{\max}}\|M_{\nu_{\max}}\|^2 - l_{\eta_{\min}}\|M_{\eta_{\min}}\|^2)^2$$

Let $\nu_{\max} = \arg\max_\nu \ell_\theta^I(s,\nu), \eta_{\min} = \arg\min_\eta \ell_\theta^I(s,\eta)$.

$$\mathbb{V}_\nu[\ell_\theta^I(s,\nu)]$$

$$\leq \frac{1}{n}\cdot((L_{\nu_{\max}} - l_{\eta_{\min}})\|M_{\nu_{\max}}\|^2 + l_{\eta_{\min}}(\|M_{\nu_{\max}}\|^2 - \|M_{\eta_{\min}}\|^2))^2$$

$$= \frac{1}{n}\cdot((L_{\nu_{\max}} - l_{\eta_{\min}})\|M_{\nu_{\max}}\|^2$$
$$+ l_{\eta_{\min}}(\|\pi_\theta(f_{\nu_{\max}}(s)) - \pi^*(s)\|^2 - \|\pi_\theta(f_{\eta_{\min}}(s)) - \pi^*(s)\|^2)^2$$

$$= \frac{1}{n}\cdot((L_{\nu_{\max}} - l_{\eta_{\min}})\|M_{\nu_{\max}}\|^2$$
$$+ l_{\eta_{\min}}(\sum_i(\pi_\theta(f_{\nu_{\max}}(s))_i - \pi^*(s)_i)^2 - \sum_i(\pi_\theta(f_{\eta_{\min}}(s))_i - \pi^*(s)_i)^2)^2$$

$$= \frac{1}{n}\cdot((L_{\nu_{\max}} - l_{\eta_{\min}})\|M_{\nu_{\max}}\|^2$$
$$+ l_{\eta_{\min}}(\sum_i\Big((\pi_\theta(f_{\nu_{\max}}(s))_i - \pi^*(s)_i)^2 - (\pi_\theta(f_{\eta_{\min}}(s))_i - \pi^*(s)_i)^2\Big))^2$$

$$\leq \frac{1}{n}\cdot((L_{\nu_{\max}} - l_{\eta_{\min}})\|M_{\nu_{\max}}\|^2$$
$$+ l_{\eta_{\min}}\sum_i\Big((\pi_\theta(f_{\nu_{\max}}(s))_i - \pi^*(s)_i)^2 - (\pi_\theta(f_{\eta_{\min}}(s))_i - \pi^*(s)_i)^2\Big)^2$$

$$= \frac{1}{n}\cdot((L_{\nu_{\max}} - l_{\eta_{\min}})\|M_{\nu_{\max}}\|^2$$
$$+ l_{\eta_{\min}}\sum_i\Big(\pi_\theta(f_{\nu_{\max}}(s))_i + \pi_\theta(f_{\eta_{\min}}(s))_i - 2\pi^*(s)_i\Big)^2\Big(\pi_\theta(f_{\nu_{\max}}(s))_i - \pi_\theta(f_{\eta_{\min}}(s))_i\Big)^2$$

$$= \frac{1}{n}\cdot((L_{\nu_{\max}} - l_{\eta_{\min}})\|M_{\nu_{\max}}\|^2$$
$$+ l_{\eta_{\min}}\sum_i\Big(\pi_\theta(f_{\nu_{\max}}(s))_i + \pi_\theta(f_{\eta_{\min}}(s))_i - 2\pi^*(s)_i\Big)^2\Big(\pi_\theta(f_{\nu_{\max}}(s))_i - \pi_\theta(f_{\eta_{\min}}(s))_i\Big)^2$$

$$= \frac{1}{n}\cdot((L_{\nu_{\max}} - l_{\eta_{\min}})\|M_{\nu_{\max}}\|^2$$
$$+ l_{\eta_{\min}}\sum_i\Big(\pi_\theta(f_{\nu_{\max}}(s))_i + \pi_\theta(f_{\eta_{\min}}(s))_i - 2\pi^*(s)_i\Big)^2\Big(\pi_\theta(f_{\nu_{\max}}(s))_i - \pi_\theta(f_{\eta_{\min}}(s))_i\Big)^2$$

$$\tag{70}$$

Since

$$
\begin{aligned}
(a - b)^2 &(a + b - 2c)^2 \\
&\leq \frac{(a-b)^4 + (a+b-2c)^4}{2} \\
&= \frac{(a-b)^4 + ((a-c+b-c)^2)^2}{2} \\
&\leq \frac{(a-b)^4 + (2(a-c)^2 + 2(b-c)^2)^2}{2} \\
&= \frac{(a-b)^4 + 4((a-c)^2 + (b-c)^2)^2}{2} \\
&\leq \frac{(a-b)^4 + 8(a-c)^4 + 8(b-c)^4}{2}
\end{aligned}
\tag{71}
$$

we have

$$
\begin{aligned}
\mathbb{V}_\nu&[\ell_\theta^I(s,\nu)] \\
&\leq \frac{1}{n} \cdot (L_{\nu_{\max}} - l_{\eta_{\min}}) \|M_{\nu_{\max}}\|^2 \\
&+ l_{\eta_{\min}} \sum_i \left(\pi_\theta(f_{\nu_{\max}}(s))_i + \pi_\theta(f_{\eta_{\min}}(s))_i - 2\pi^*(s)_i\right)^2 \left(\pi_\theta(f_{\nu_{\max}}(s))_i - \pi_\theta(f_{\eta_{\min}}(s))_i\right)^2 \\
&\leq \frac{1}{n} \cdot (L_{\nu_{\max}} - l_{\eta_{\min}}) \|M_{\nu_{\max}}\|^2 \\
&+ \frac{1}{2} l_{\eta_{\min}} \|\pi_\theta(f_{\nu_{\max}}(s)) - \pi_\theta(f_{\eta_{\min}}(s))\|^4 \\
&+ 4 l_{\eta_{\min}} \|\pi_\theta(f_{\nu_{\max}}(s)) - \pi^*(s)\|^4 \\
&+ 4 l_{\eta_{\min}} \|\pi_\theta(f_{\eta_{\min}}(s)) - \pi^*(s)\|^4
\end{aligned}
\tag{72}
$$

$\square$

In Equation 72, the third and fourth terms are minimized by the actor loss. If we minimize the second term of Equation 72 by minimizing the mean squared error between two deterministic actions $||\pi_\theta(f_\eta(s')) - \pi_\theta(f_\nu(s'))||^2$ in the case of DDPG, the variance of the actor loss is reduced.

### E.3 KL REDUCES THE VARIANCE OF THE TARGET Q-VALUE

**DDPG target values** For DDPG, when we compute target values, we add Ornstein-Uhlenbeck noise to deterministic actions for exploration. Then the policy can be regarded as a probability distribution $\pi$.

For image-based control tasks, a data augmentation $f$ parameterized by $\mu \sim \mathcal{P}$ is applied on the observations. Then the target value $y$ for a given transition $(s, a, r, s')$ is

$$
y(f_\mu(s'), a') = r + \gamma Q_{\bar\phi}(f_\mu(s'), a'), \text{where } a' \sim \pi(\cdot|f_\mu(s')).
\tag{73}
$$

The expectation of $y(f_\mu(s'), a')$ with respect to $a' \sim \pi(\cdot|f_\mu(s'))$ is

$$
\begin{aligned}
\mathbb{E}_{a'}[y(f_\mu(s'), a')] &= r + \gamma \mathbb{E}_{a'}[Q_{\bar\phi}(f_\mu(s'), a')] \\
&= r + \gamma \sum_{a'} \pi(a'|f_\mu(s')) Q_{\bar\phi}(f_\mu(s'), a')
\end{aligned}
\tag{74}
$$

The expectation of $y(f_\mu(s'), a')$ with respect to $\mu \sim \mathcal{P}$ and $a' \sim \pi(\cdot|f_\mu(s'))$ is

$$
\begin{aligned}
\mathbb{E}_{\mu,a'}&[y(f_\mu(s'), a')] \\
&= r + \gamma \mathbb{E}_{\mu,a'}[Q_{\bar\phi}(f_\mu(s'), a')] \\
&= r + \gamma \sum_\mu \mathcal{P}(\mu) \sum_{a'} \pi(a'|f_\mu(s')) Q_{\bar\phi}(f_\mu(s'), a')
\end{aligned}
\tag{75}
$$

We create two tables to better illustrate the meanings of $\mathbb{E}_{a'}[y(f_\mu(s'), a')]$ and $\mathbb{E}_{\mu,a'}[y(f_\mu(s'), a')]$.

| $a'$ \ $\mu$ | ... | $f_{\tau_m}(s')$ with $\mathcal{P}(\mu = \tau_m)$ | ... |
|---|---|---|---|
| $a_1'$ | ... | $y(f_{\tau_m}(s'), a_1')$ with $\mathcal{P}(\mu = \tau_m) \cdot \pi_\theta(a_1'\|f_{\tau_m}(s'))$ | ... |
| $a_2'$ | ... | $y(f_{\tau_m}(s'), a_2')$ with $\mathcal{P}(\mu = \tau_m) \cdot \pi_\theta(a_2'\|f_{\tau_m}(s'))$ | ... |
| ... | ... | ... | ... |
| $a_n'$ | ... | $y(f_{\tau_m}(s'), a_n')$ with $\mathcal{P}(\mu = \tau_m) \cdot \pi_\theta(a_n'\|f_{\tau_m}(s'))$ | ... |
| ... | ... | ... | ... |
| $\mathbb{E}[y]$ wrt. $a'$ | ... | $\mathbb{E}_{a'}[y(f_{\tau_m}(s'), a')]$ | ... |

| $f_{\tau_1}(s')$ with $\mathcal{P}(\mu = \tau_1)$ | ... | $\mathbb{E}[y]$ wrt. $a$ and $\mu$ |
|---|---|---|
| $\mathbb{E}_{a'}[y(f_{\tau_1}(s'), a')]$ | ... | $\mathbb{E}_{\mu,a'}[y(f_\mu(s'), a')]$ |

The variance of $y(f_\mu(s'), a')$ with respect to $\mu$ and $a'$ is

$$\mathbb{V}_{\mu,a'}[y(f_\mu(s'), a')]$$
$$= \sum_\mu \mathcal{P}(\mu) \sum_{a'} \pi(a'|f_\mu(s')) \Big[ (y(f_\mu(s'), a') - \mathbb{E}_{\mu,a'}[y(f_\mu(s'), a')])^2 \Big]$$
$$= \sum_\mu \mathcal{P}(\mu) \sum_{a'} \pi(a'|f_\mu(s'))$$
$$\Big[ (y(f_\mu(s'), a') - \mathbb{E}_{a'}[y(f_\mu(s'), a')] + \mathbb{E}_{a'}[y(f_\mu(s'), a')] - \mathbb{E}_{\mu,a'}[y(f_\mu(s'), a')])^2 \Big] \quad (76)$$
$$= \sum_\mu \mathcal{P}(\mu) \sum_{a'} \pi(a'|f_\mu(s')) \Big[ (y(f_\mu(s'), a') - \mathbb{E}_{a'}[y(f_\mu(s'), a')])^2$$
$$+ 2(y(f_\mu(s'), a') - \mathbb{E}_{a'}[y(f_\mu(s'), a')]) \cdot (\hat{\mathbb{E}}_{a'}[y(f_\mu(s'), a')] - \mathbb{E}_{\mu,a'}[y(f_\mu(s'), a')])$$
$$+ (\mathbb{E}_{a'}[y(f_\mu(s'), a')] - \mathbb{E}_{\mu,a'}[y(f_\mu(s'), a')])^2 \Big]$$

The first term of Equation 76 is the expectation of squared advantage.

The second term of Equation 76 is 0 because

$$\sum_\mu \mathcal{P}(\mu) \sum_{a'} \pi(a'|f_\mu(s')) \Big[ 2(y(f_\mu(s'), a') - \mathbb{E}_{a'}[y(f_\mu(s'), a')])$$
$$\cdot (\mathbb{E}_{a'}[y(f_\mu(s'), a')] - \mathbb{E}_{\mu,a'}[y(f_\mu(s'), a')]) \Big]$$
$$= 2 \sum_\mu \Big[ \mathcal{P}(\mu) \cdot (\mathbb{E}_{a'}[y(f_\mu(s'), a')] - \mathbb{E}_{\mu,a'}[y(f_\mu(s'), a')])$$
$$\cdot \Big( \sum_{a'} \pi(a'|f_\mu(s'))(y(f_\mu(s'), a') - \mathbb{E}_{a'}[y(f_\mu(s'), a')]) \Big) \Big] \quad (77)$$
$$= 2 \sum_\mu \Big[ \mathcal{P}(\mu) \cdot (\mathbb{E}_{a'}[y(f_\mu(s'), a')] - \mathbb{E}_{\mu,a'}[y(f_\mu(s'), a')])$$
$$\cdot \Big( \mathbb{E}_{a'}[y(f_\mu(s'), a')] - \mathbb{E}_{a'}[y(f_\mu(s'), a')]) \Big) \Big]$$
$$= 0$$

The third term of Equation 76 is the variance of $\mathbb{E}_{a' \sim \pi(\cdot|f_\mu(s'))}[y(f_\mu(s'), a')]$ with respect to $\mu$. Both the variance $\mathbb{V}_\mu[\mathbb{E}_{a' \sim \pi(\cdot|f_\mu(s'))}[y(f_\mu(s'), a')]]$ and the variance of the empirical mean

$\mathbb{V}_\mu[\hat{\mathbb{E}}_\mu[\mathbb{E}_{a'\sim\pi_\theta(\cdot|f_\mu(s'))}[y(f_\mu(s'),a')]]]$ are bounded by the KL divergence $D_{\eta,\mu} = D_{KL}(\pi(\cdot \mid f_\eta(s')) \mid \pi(\cdot \mid f_\mu(s')))$ for $\mu, \eta \sim \mathcal{P}$ if $Q_{\bar\phi}(f_\mu(s'),a')$ is invariant with respect to $\mu$ for all $a' \in \mathcal{A}$.

*Proof.*

$$\mathbb{V}_\mu[\mathbb{E}_{a'\sim\pi(\cdot|f_\mu(s'))}[y(f_\mu(s'),a')]] = \mathbb{E}_\mu\Big[(\mathbb{E}_{a'}[y(f_\mu(s'),a')] - \mathbb{E}_{\eta,a'}[y(f_\eta(s'),a')])^2\Big] \tag{78}$$

$$
\begin{aligned}
&\mathbb{E}_{a'}[y(f_\mu(s'),a')] - \mathbb{E}_{\eta,a'}[y(f_\eta(s'),a')] \\
&= \gamma\Big((\sum_{a'}\pi(a'|f_\mu(s'))Q_{\bar\phi}(f_\mu(s'),a')) - (\sum_\eta \mathcal{P}(\eta)\int_{a'}\pi(a'|f_\eta(s'))Q_{\bar\phi}(f_\eta(s'),a'))\Big) \\
&= \gamma\Big(((\sum_\eta\mathcal{P}(\eta)\sum_{a'}\pi(a'|f_\mu(s'))Q_{\bar\phi}(f_\mu(s'),a')) - (\sum_\eta\mathcal{P}(\eta)\sum_{a'}\pi(a'|f_\eta(s'))Q_{\bar\phi}(f_\eta(s'),a')))\Big) \\
&= \gamma\Big(\sum_\eta\mathcal{P}(\eta)\sum_{a'}\pi(a'|f_\mu(s'))Q_{\bar\phi}(f_\mu(s'),a') - \pi(a'|f_\eta(s'))Q_{\bar\phi}(f_\eta(s'),a')\Big) \\
&= \gamma\Big(\sum_\eta\mathcal{P}(\eta)\sum_{a'}\pi(a'|f_\mu(s'))Q_{\bar\phi}(f_\mu(s'),a') - \pi(a'|f_\eta(s'))Q_{\bar\phi}(f_\mu(s'),a') \\
&\quad + \pi(a'|f_\eta(s'))Q_{\bar\phi}(f_\mu(s'),a') - \pi(a'|f_\eta(s'))Q_{\bar\phi}(f_\eta(s'),a')\Big) \\
&= \gamma\Big(\sum_\eta\mathcal{P}(\eta)\sum_{a'}(\pi(a'|f_\mu(s')) - \pi(a'|f_\eta(s')))Q_{\bar\phi}(f_\mu(s'),a') \\
&\quad + \pi(a'|f_\eta(s'))(Q_{\bar\phi}(f_\mu(s'),a') - Q_{\bar\phi}(f_\eta(s'),a'))\Big)
\end{aligned}
\tag{79}
$$

The second term of Equation 79 $\gamma\sum_\eta\mathcal{P}(\eta)\sum_{a'}\pi(a'|f_\eta(s'))(Q_{\bar\phi}(f_\mu(s'),a') - Q_{\bar\phi}(f_\eta(s'),a'))$ is related to the difference of $Q_{\bar\phi}(f_\eta(s'),a')$ and $Q_{\bar\phi}(f_\mu(s'),a')$, which is governed by the critic loss. When $Q_{\bar\phi}(f_\mu(s'),a')$ is invariant with respect to $\mu$ for all $a' \in \mathcal{A}$, this term is zero.

For the first term of Equation 79,

$$
\begin{aligned}
&\sum_\eta\mathcal{P}(\eta)\sum_{a'}(\pi(a'|f_\mu(s')) - \pi(a'|f_\eta(s')))Q_{\bar\phi}(f_\mu(s'),a') \\
&\leq \sum_\eta\mathcal{P}(\eta)\sum_{a'}|\pi(a'|f_\mu(s')) - \pi(a'|f_\eta(s'))|Q_{\bar\phi}(f_\mu(s'),a') \\
&\leq \max_{a'}Q_{\bar\phi}(f_\mu(s'),a')\cdot\sum_\eta\mathcal{P}(\eta)\sum_{a'}|\pi(a'|f_\mu(s')) - \pi(a'|f_\eta(s'))| \\
&\leq \max_{a'}Q_{\bar\phi}(f_\mu(s'),a')\cdot\sum_\eta\mathcal{P}(\eta)\sqrt{2D_{KL}(\pi(\cdot|f_\eta(s'))||\pi(\cdot|f_\mu(s')))}
\end{aligned}
\tag{80}
$$

where in the first inequality absolute values $|\pi(a'|f_\mu(s')) - \pi(a'|f_\eta(s'))|$ are applied, in the second inequality $Q_{\bar\phi}(f_\mu(s'),a')$ is replaced with $\max_{a'}Q_{\bar\phi}(f_\mu(s'),a')$ and Pinsker's inequality is applied in the third inequality.

Similarly, a lower bound can be derived.

$$
\begin{aligned}
&\sum_\eta\mathcal{P}(\eta)\sum_{a'}(\pi(a'|f_\mu(s')) - \pi(a'|f_\eta(s')))Q_{\bar\phi}(f_\mu(s'),a') \\
&\geq -\max_{a'}Q_{\bar\phi}(f_\mu(s'),a')\cdot\sum_\eta\mathcal{P}(\eta)\sqrt{2D_{KL}(\pi(\cdot|f_\eta(s'))||\pi(\cdot|f_\mu(s')))}
\end{aligned}
\tag{81}
$$

Therfore,

$$\mathbb{V}_\mu[\mathbb{E}_{a'\sim\pi(\cdot|f_\mu(s'))}[y(f_\mu(s'),a')]] \leq \mathbb{E}_\mu\Big[\gamma^2\Big(\max_{a'}Q_{\bar\phi}(f_\mu(s'),a')\mathbb{E}_\eta\Big[\sqrt{2D_{\eta,\mu}}\Big]\Big)^2\Big] \tag{82}$$

Let $\hat{Y}(s', \mu) = \hat{\mathbb{E}}_\mu[\mathbb{E}_{a' \sim \pi_\theta(\cdot|f_\mu(s'))}[y(f_\mu(s'), a')]]$.

From Equation 61,

$$\mathbb{V}_\mu[\hat{Y}(s', \mu)] = \frac{1}{n} \mathbb{V}_\mu[\mathbb{E}_{a' \sim \pi_\theta(\cdot|f_\mu(s'))}[y(f_\mu(s'), a')]], \tag{83}$$

where $n$ is the number of samples to estimate the empirical mean $\hat{Y}(s', \mu)$.

Therefore, if $Q_{\bar{\phi}}(f_\mu(s), a')$ is invariant with respect to $\mu$ for all $a' \in \mathcal{A}$, the variance of $\hat{Y}(s', \mu)$ with respect to $\mu$ is bounded by the KL divergence $D_{\eta,\mu} = D_{KL}(\pi(\cdot \mid f_\eta(s')) \mid \pi(\cdot \mid f_\mu(s')))$ for $\mu, \eta \sim \mathcal{P}$.

$$\mathbb{V}_\mu[\hat{Y}(s', \mu)] \leq \frac{1}{n} \mathbb{E}_\mu\left[\gamma^2 \left(\max_{a'} Q_{\bar{\phi}}(f_\mu(s'), a') \mathbb{E}_\eta\left[\sqrt{2D_{\eta,\mu}}\right]\right)^2\right] \tag{84}$$

For DDPG, minimizing the KL divergence between policy distributions of two augmented states $D_{KL}(\pi(\cdot \mid f_\eta(s')) \mid \pi(\cdot \mid f_\mu(s')))$ is equivalent to minimizing the mean squared error between two deterministic actions $||\bar{\pi}(f_\eta(s')) - \bar{\pi}(f_\mu(s'))||^2$. $\qquad\square$

**SAC target value with the entropy term** If the entropy term is added to the target value, the variance of the empirical mean $\mathbb{V}_\mu[\hat{\mathbb{E}}_\mu[\mathbb{E}_{a' \sim \pi_\theta(\cdot|f_\mu(s'))}[y(f_\mu(s'), a')]]]$ is still bounded by the KL divergence $D_{\eta,\mu} = D_{KL}(\pi(\cdot \mid f_\eta(s')) \mid \pi(\cdot \mid f_\mu(s')))$ for $\mu, \eta \sim \mathcal{P}$ if $Q_{\bar{\phi}}(f_\mu(s'), a')$ is invariant with respect to $\mu$ for all $a' \in \mathcal{A}$.

$$\mathbb{V}_\mu[\hat{\mathbb{E}}_\mu[\mathbb{E}_{a' \sim \pi_\theta(\cdot|f_\mu(s'))}[y(f_\mu(s'), a')]]] \leq \frac{1}{n} \mathbb{E}_\mu\left[\left(\mathbb{E}_\eta\left[\max_{a'}(y(f_\mu(s'), a') - r)\sqrt{2D_{\eta,\mu}} + \alpha \cdot D_{\eta,\mu}\right]\right)^2\right] \tag{85}$$

where $n$ is the number of samples to estimate the empirical mean, $r$ is the reward of this transition and $\alpha$ is the entropy coefficient.

*Proof.* After we add the entropy term, the target value becomes

$$y(f_\mu(s'), a') = r + \gamma Q_{\bar{\phi}}(f_\mu(s'), a') - \alpha \log \pi(a'|f_\mu(s')), \tag{86}$$

where $a' \sim \pi(\cdot|f_\mu(s'))$ and $\alpha$ is the entropy coefficient.

Let

$$y_1(f_\mu(s'), a') = y(f_\mu(s'), a') - r = \gamma Q_{\bar{\phi}}(f_\mu(s'), a') - \alpha \log \pi(a'|f_\mu(s')) \tag{87}$$

Since $r$ is a constant value, we can drop $r$ when calculating the variance.

$$\begin{aligned}
&\mathbb{V}_\mu[\mathbb{E}_{a' \sim \pi(\cdot|f_\mu(s'))}[y(f_\mu(s'), a')]] \\
&= \mathbb{V}_\mu[\mathbb{E}_{a' \sim \pi(\cdot|f_\mu(s'))}[y_1(f_\mu(s'), a')]] \\
&= \mathbb{E}_\mu\left[(\mathbb{E}_{a'}[\gamma Q_{\bar{\phi}}(f_\mu(s'), a') - \alpha \log \pi(a'|f_\mu(s'))] - \mathbb{E}_{\eta,a'}[\gamma Q_{\bar{\phi}}(f_\eta(s'), a') - \alpha \log \pi(a'|f_\eta(s'))])^2\right]
\end{aligned} \tag{88}$$

$$\mathbb{E}_{a'}[y_1(f_\mu(s'), a')] - \mathbb{E}_{\eta, a'}[y_1(f_\eta(s'), a')]$$

$$= \sum_{a'} \pi(a'|f_\mu(s'))y_1(f_\mu(s'), a') - \sum_\eta \mathcal{P}(\eta) \int_{a'} \pi(a'|f_\eta(s'))y_1(f_\eta(s'), a')$$

$$= \sum_\eta \mathcal{P}(\eta) \sum_{a'} \pi(a'|f_\mu(s'))y_1(f_\mu(s'), a') - \sum_\eta \mathcal{P}(\eta) \sum_{a'} \pi(a'|f_\eta(s'))y_1(f_\eta(s'), a')$$

$$= \sum_\eta \mathcal{P}(\eta) \sum_{a'} \pi(a'|f_\mu(s'))y_1(f_\mu(s'), a') - \pi(a'|f_\eta(s'))y_1(f_\eta(s'), a')$$

$$= \sum_\eta \mathcal{P}(\eta) \sum_{a'} \pi(a'|f_\mu(s'))y_1(f_\mu(s'), a') - \pi(a'|f_\eta(s'))y_1(f_\mu(s'), a')$$

$$+ \pi(a'|f_\eta(s'))y_1(f_\mu(s'), a') - \pi(a'|f_\eta(s'))y_1(f_\eta(s'), a')$$

$$= \sum_\eta \mathcal{P}(\eta) \sum_{a'} y_1(f_\mu(s'), a')(\pi(a'|f_\mu(s')) - \pi(a'|f_\eta(s')))$$

$$+ \pi(a'|f_\eta(s'))(y_1(f_\mu(s'), a') - y_1(f_\eta(s'), a'))$$

$$= \sum_\eta \mathcal{P}(\eta) \sum_{a'} y_1(f_\mu(s'), a')(\pi(a'|f_\mu(s')) - \pi(a'|f_\eta(s'))) \tag{89}$$

$$+ \pi(a'|f_\eta(s'))(\gamma Q_{\bar\phi}(f_\mu(s'), a') - \gamma Q_{\bar\phi}(f_\eta(s'), a'))$$

$$+ \pi(a'|f_\eta(s'))(\alpha \log \pi(a'|f_\eta(s')) - \alpha \log \pi(a'|f_\mu(s')))$$

$$= \sum_\eta \mathcal{P}(\eta) \sum_{a'} y_1(f_\mu(s'), a')(\pi(a'|f_\mu(s')) - \pi(a'|f_\eta(s')))$$

$$+ \pi(a'|f_\eta(s'))(\gamma Q_{\bar\phi}(f_\mu(s'), a') - \gamma Q_{\bar\phi}(f_\eta(s'), a'))$$

$$+ \sum_\eta \mathcal{P}(\eta) \sum_{a'} \pi(a'|f_\eta(s'))(\alpha \log \pi(a'|f_\eta(s')) - \alpha \log \pi(a'|f_\mu(s')))$$

$$= \sum_\eta \mathcal{P}(\eta) \sum_{a'} y_1(f_\mu(s'), a')(\pi(a'|f_\mu(s')) - \pi(a'|f_\eta(s')))$$

$$+ \gamma \sum_\eta \mathcal{P}(\eta) \sum_{a'} \pi(a'|f_\eta(s'))(Q_{\bar\phi}(f_\mu(s'), a') - Q_{\bar\phi}(f_\eta(s'), a'))$$

$$+ \sum_\eta \mathcal{P}(\eta)\alpha \cdot D_{KL}(\pi(a'|f_\eta(s'))|\pi(a'|f_\mu(s')))$$

Similar to Equation 80 and Equation 81, we apply Pinsker's inequality and obtain the lower and the upper bounds for the first term of Equation 89.

$$- \max_{a'} y_1(f_\mu(s'), a') \cdot \sum_\eta \mathcal{P}(\eta)\sqrt{2D_{KL}(\pi(\cdot|f_\eta(s'))||\pi(\cdot|f_\mu(s')))}$$

$$\leq \sum_\eta \mathcal{P}(\eta) \sum_{a'}(\pi(a'|f_\mu(s')) - \pi(a'|f_\eta(s')))y(f_\mu(s'), a') \tag{90}$$

$$\leq \max_{a'} y_1(f_\mu(s'), a') \cdot \sum_\eta \mathcal{P}(\eta)\sqrt{2D_{KL}(\pi(\cdot|f_\eta(s'))||\pi(\cdot|f_\mu(s')))}$$

The second term of Equation 89 $\gamma \sum_\eta \mathcal{P}(\eta) \sum_{a'} \pi(a'|f_\eta(s'))(Q_{\bar\phi}(f_\mu(s'), a') - Q_{\bar\phi}(f_\eta(s'), a'))$ is related to the difference of $Q_{\bar\phi}(f_\eta(s'), a')$ and $Q_{\bar\phi}(f_\mu(s'), a')$, which is governed by the critic loss. When $Q_{\bar\phi}(f_\mu(s'), a')$ is invariant with respect to $\mu$ for all $a' \in \mathcal{A}$, this term is zero. $\qquad\square$

Therefore,

$$\sum_{\eta} \mathcal{P}(\eta) \cdot \Big( - \max_{a'} y_1(f_\mu(s'), a') \cdot \sqrt{2D_{KL}(\pi(\cdot|f_\eta(s'))||\pi(\cdot|f_\mu(s')))} + \alpha \cdot D_{KL}(\pi(a'|f_\eta(s'))|\pi(a'|f_\mu(s'))) \Big)$$

$$\leq \mathbb{E}_{a'}[y(f_\mu(s'), a')] - \mathbb{E}_{\eta, a'}[y(f_\eta(s'), a')]$$

$$\leq \sum_{\eta} \mathcal{P}(\eta) \cdot \Big( \max_{a'} y_1(f_\mu(s'), a') \cdot \sqrt{2D_{KL}(\pi(\cdot|f_\eta(s'))||\pi(\cdot|f_\mu(s')))} + \alpha \cdot D_{KL}(\pi(a'|f_\eta(s'))|\pi(a'|f_\mu(s'))) \Big)$$

$$(91)$$

Plug the above inequalities into Equation 88, we obtain

$$\mathbb{V}_\mu[\mathbb{E}_{a' \sim \pi(\cdot|f_\mu(s'))}[y(f_\mu(s'), a')]] \leq \mathbb{E}_\mu \Big[ \Big( \mathbb{E}_\eta \Big[ \max_{a'} y_1(f_\mu(s'), a') \sqrt{2D_{\eta,\mu}} + \alpha \cdot D_{\eta,\mu} \Big] \Big)^2 \Big] \quad (92)$$

If $Q_{\bar{\phi}}(f_\mu(s), a')$ is invariant with respect to $\mu$ for all $a' \in \mathcal{A}$, the variance of $\hat{Y}(s', \mu)$ with respect to $\mu$ is bounded by the KL divergence $D_{\eta,\mu} = D_{KL}(\pi(\cdot \mid f_\eta(s')) \mid \pi(\cdot \mid f_\mu(s')))$ for $\mu, \eta \sim \mathcal{P}$.

$$\mathbb{V}_\mu[\hat{Y}(s', \mu)] \leq \frac{1}{n} \mathbb{E}_\mu \Big[ \Big( \mathbb{E}_\eta \Big[ \max_{a'} (y(f_\mu(s'), a') - r) \sqrt{2D_{\eta,\mu}} + \alpha \cdot D_{\eta,\mu} \Big] \Big)^2 \Big] \quad (93)$$

## F  CALCULATING TARGET WITH COMPLEX DATA AUGMENTATION

In this section, we experimentally analyze using complex image transformations in calculating the target and show that cosine similarity of the augmented features at the early training stage can be used as a criteria for judging if an image transformation is complex or not. Sufficient updates is the key condition for good performance when using complex image transformations in calculating the target.

In contrast to the analysis in SVEA (Hansen et al., 2021), we observe that even using complex image transformation such as random conv in the target does not induce a large variance in the target. Instead, a much larger bias is observed for the trained agent, as shown in the Table 5. This can be solved by increasing the number of updates, as shown in Figure 4.

Furthermore, we test with other image transformations which are regarded as complex image transformations in SVEA (Hansen et al., 2021). In order to show whether it's easy to enforce the invariance of a image transformation, we record the cosine similarities of encoder outputs for two augmented images transformed by this image transformation, as shown in Table 4. For image transformations such as random overlay or gaussian blur, the invariance is easy to enforce and the cosine similarities are large. When using this kind of image transformation in calculating the target values, it won't hurt the performance. Otherwise, for image transformations such as random convolution or random rotation, the invariance is relatively harder to enforce during training and the cosine similarities are small. Then directly applying this kind of image transformations in calculating the target values will decrease the learning efficiency. To resolve this issue, we need more updates for each training step. The evaluation results for SVEA with random overlay, random convolution, random rotation and gaussian blur are shown in Figure 5.

## G  HYPERPARAMETERS

Hyperparameters used in experiments on DMControl (drq), DMControl (drqv2) and DMGB can be found in Table 6, Table 7 and 8. For experiments in DMControl(drqv2) and DMGB, when applicable, we adopt hyperparameters from the official implementation of drqv2 by Yarats et al. (2021) and SVEA by Hansen et al. (2021) respectively.

## H  ADDITIONAL RESULTS

### H.1  ABLATION STUDY

The performance profile is shown in Figure 6 and the training curves in different environments are shown in Figure 7.

| Statistics | svea(DA=blur) | svea(DA=overlay) | svea(DA=conv) | svea(DA=rotation) |
|---|---|---|---|---|
| actor sim (shift) | 0.919±0.007 | 0.911±0.005 | 0.910±0.011 | 0.936±0.006 |
| actor sim (DA) | 0.998±0.001 | 0.906±0.006 | 0.854±0.019 | 0.536±0.069 |
| critic sim (shift) | 0.938±0.010 | 0.942±0.003 | 0.922±0.010 | 0.962±0.005 |
| critic sim (DA) | 0.998±0.001 | 0.939±0.003 | 0.883±0.014 | 0.660±0.057 |

Table 4: Recorded cosine similarity for latent features at 100k steps in walker walk environment for SVEA trained with different complex image transformations. Here, each column corresponds to SVEA with different image transformations and each row corresponds to a cosine similarity recorded at 100k steps. For example, the first number is calculated by the cosine similarity between the latent features $E_\pi(f_{shift}(s))$, in which $E_\pi$ is the encoder of the actor in SVEA trained with gaussian blur and $f_{shift}$ is the random shift. Considering that random shift is always applied in SVEA, the cosine similarity with respect to it is recorded as the baseline for comparisons. For gaussian blur and random overlay (second and third column), the cosine similarities of latent features are higher or similar to the cosine similarities between the latent features from two randomly shifted images which means they are not complex image transformations. In contrast, random conv and random rotation (last two columns) leads to smaller cosine similarities of latent features which indicates that they are relatively complex image transformations in this environment.

| Statistics | Step 100k | Step 200k | Step 300k | Step 400k | Step 500k |
|---|---|---|---|---|---|
| Mean (w/ conv) | 64.05 | 112.44 | 140.78 | 166.42 | 185.92 |
| Mean (w/o conv) | 84.96 | 148.46 | 197.10 | 221.67 | 239.35 |
| Variance (w/ conv) | 1.228 | 1.109 | 1.148 | 1.323 | 1.306 |
| Variance (w/o conv) | 0.882 | 0.812 | 0.885 | 0.757 | 0.795 |
| Bias (w/ conv) | 52.88 | 67.44 | 54.20 | 63.89 | 55.36 |
| Bias (w/o conv) | 77.40 | 60.48 | 50.40 | 43.22 | 28.75 |

Table 5: Mean, variance and bias of the target Q-values for the agent trained with/without using random conv in calculating the target. Here, the mean and variance are calculated by the mean and variance of a set of sampled target Q values and the bias is calculated by the mean-squared error between the targets used in the training and the true targets estimated by the sum of discounted rewards from sampled trajectories. At the end of the training, the increase in the variance when using random conv is not significant compared to the mean of the target. However, the bias is much larger at the end.

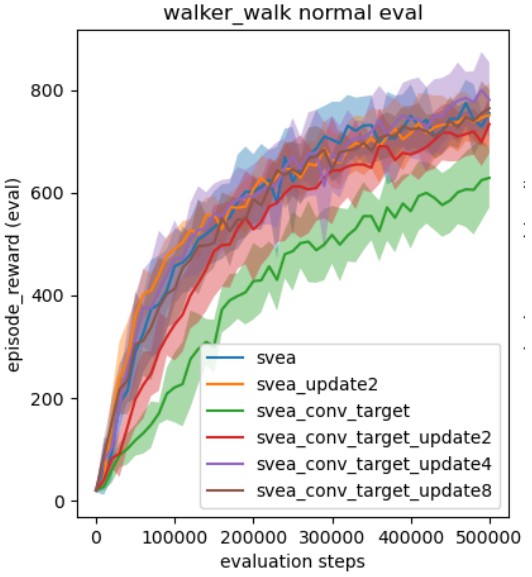

Figure 4: Performance of increasing the number of updates with/without using random conv in calculating the targets.

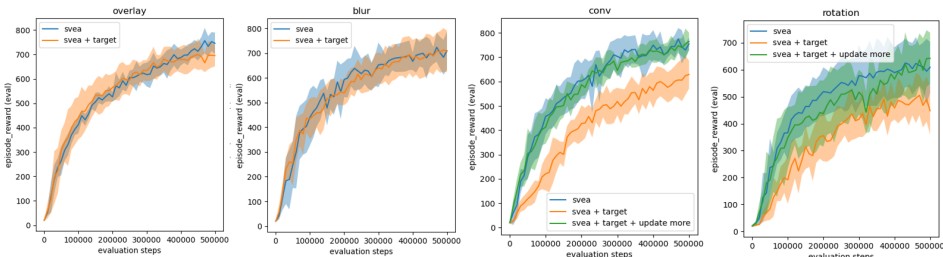

Figure 5: Performance of increasing the number of updates in walker walk environment when using complex image transformation in calculating the targets. For random convolution and random rotation, "update more" stands for doing 4 updates for each training step.

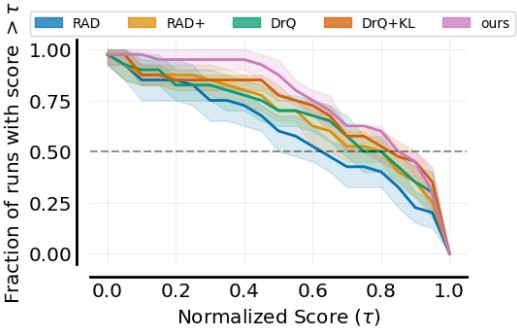

Figure 6: Performance profile of different methods.

Table 6: Hyperparameters used in experiments on DMControl (drq)

| Hyperparameter | Value on DMControl |
| --- | --- |
| frame rendering | $84 \times 84 \times 3$ |
| stacked frames | 3 |
| action repeat | 2 |
| replay buffer capacity | 100,000 |
| seed steps | 1000 |
| environment steps | 250,000 in reacher easy
250,000 in finger spin
250,000 in ball
500,000 in others |
| batch size $N$ | 256 |
| discount $\gamma$ | 0.99 |
| optimizer $(\phi, \theta)$ | Adam
$(\beta_1 = 0.9, \beta_2 = 0.999)$ |
| optimizer ($\alpha$ of SAC) | Adam
$(\beta_1 = 0.9, \beta_2 = 0.999)$ |
| learning rate $(\phi, \theta)$ | 1e-3 |
| learning rate ($\alpha$ of SAC) | 1e-3 |
| target network update frequency | 2 |
| target network soft-update rate | 0.01 |
| actor update frequency $\kappa$ | 2 |
| actor log stddev bounds | [-10,2] |
| init temperature $\alpha$ | 0.1 |
| tangent prop weight $\alpha_{tp}$ | 0.1 |
| actor KL weight $\alpha_{KL}$ | 0.1 |

Table 7: Hyperparameters used in experiments on DMControl (drqv2)

| Hyperparameter | Value on DMC |
| --- | --- |
| frame rendering | $84 \times 84 \times 3$ |
| stacked frames | 3 |
| action repeat | 2 |
| replay buffer capacity | $10^6$ |
| seed frames | 4000 |
| exploration steps | 2000 |
| n-step returns | 3 |
| batch size $N$ | 256 |
| discount $\gamma$ | 0.99 |
| optimizer $(\phi, \theta)$ | Adam |
| learning rate $(\phi, \theta)$ | 1e-4 |
| agent update frequency | 2 |
| target network soft-update rate | 0.01 |
| exploration stddev clip | 0.3 |
| exploration stddev schedule | linear(1.0, 0.1, 500000) |
| tangent prop weight $\alpha_{tp}$ | 0.1 |
| actor KL weight $\alpha_{KL}$ | 0.1 |

## H.2 CASE STUDY

The measures for invariance in the latent space are shown in Figure 8.

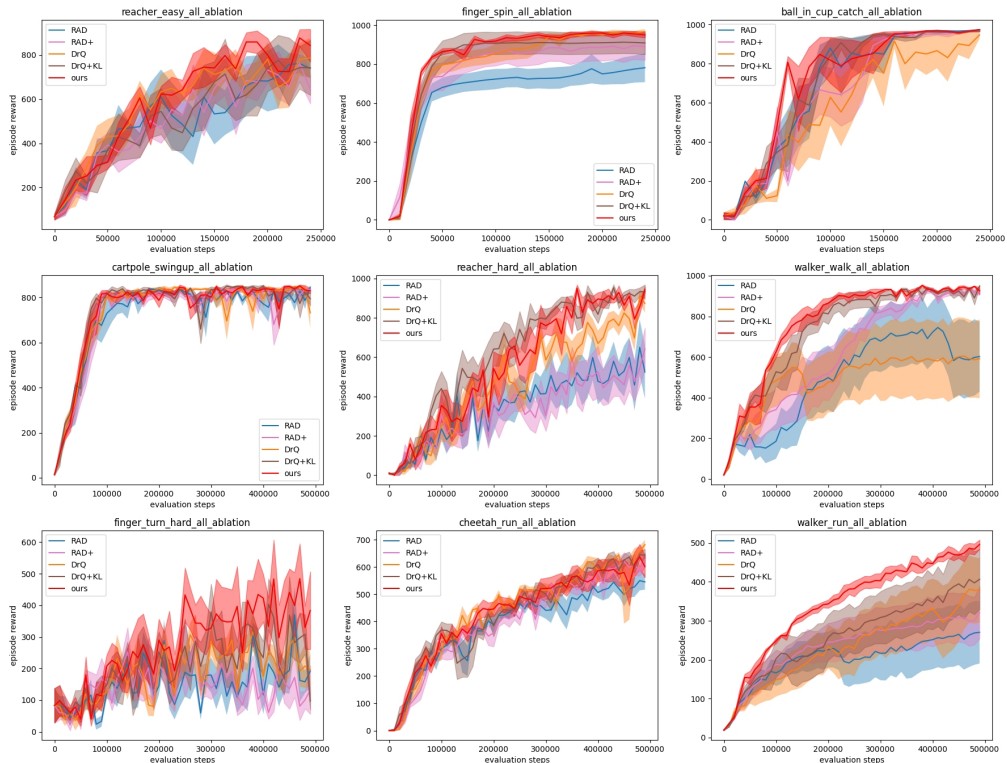

Figure 7: Full results of validating our propositions.

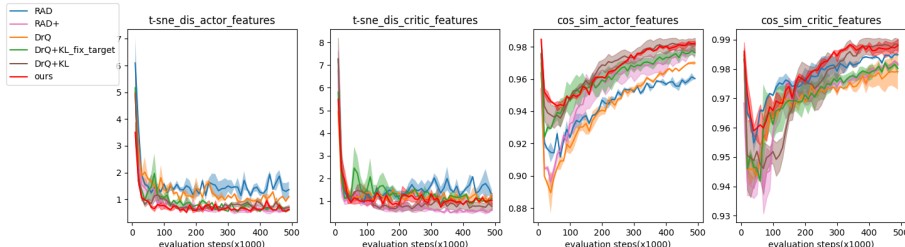

Figure 8: The figure shows the learned invariance in the feature space of the actor and critic. Two measures of the invariance are provided in this figure: the distances between projected points of the augmented features by t-SNE and the cosine similarities between augmented features.

Table 8: Hyperparameters used in experiments on DMControl Generalization Benchmark (DMGB)

| Hyperparameter | Value on DMGB |
|---|---|
| frame rendering | $84 \times 84 \times 3$ |
| stacked frames | 3 |
| action repeat | 2(finger) 8(cartpole) 4(otherwise) |
| replay buffer capacity | 500,000 / action repeat |
| seed steps | 1000 |
| environment steps | 500,000 |
| batch size $N$ | 128 |
| discount $\gamma$ | 0.99 |
| optimizer $(\phi, \theta)$ | Adam $(\beta_1 = 0.9, \beta_2 = 0.999)$ |
| optimizer ($\alpha$ of SAC) | Adam $(\beta_1 = 0.5, \beta_2 = 0.999)$ |
| learning rate $(\phi, \theta)$ | 1e-3 |
| learning rate ($\alpha$ of SAC) | 1e-4 |
| target network update frequency | 2 |
| target network soft-update rate | 0.01(critic) 0.05(encoder) |
| actor update frequency $\kappa$ | 2 |
| actor log stddev bounds | [-10,2] |
| init temperature $\alpha$ | 0.1 |
| tangent prop weight $\alpha_{tp}$ | 0.5 |
| actor KL weight $\alpha_{KL}$ | 0.1 |

### H.3 MORE EVALUATIONS

Here, we include more evaluations of our proposition. The results of comparing our proposition with DrQ are shown in Figure 9. The results of comparing our proposition with DrQv2 are shown in Figure 10.

### H.4 RESULTS OF GENERALIZATION ABILITY IN DMCONTROL GENERALIZATION BENCHMARK (DMGB)

The comparison of generalization performance in DMGB between SVEA and our method using random overlay as data augmentation is shown in Figure 11.

### H.5 RESULTS OF RECORDED STATISTICS

The curves for the recorded statistics, including standard deviation of the empirical critic loss, standard deviation of the target Q-values, and empirical mean of KL divergence between policies for two augmented samples along the training are shown in Figure 12 and Figure 13.

## I LIMITATIONS

We try to provide some recommendations on how to apply theoretically-sound data augmentation method in DRL. However, the analysis can still be further refined to be more comprehensive such as including the theoretical analysis of using different distributions for the image transformation and providing a thorough analysis on tangent prop regularization. Moreover, our method naturally requires the knowledge of some effective image transformations for a given task. Without such knowledge, the invariant transformations for a problem would need to be learned, which is currently an active research direction. Finally, image transformation may rely on some implicit assumptions,

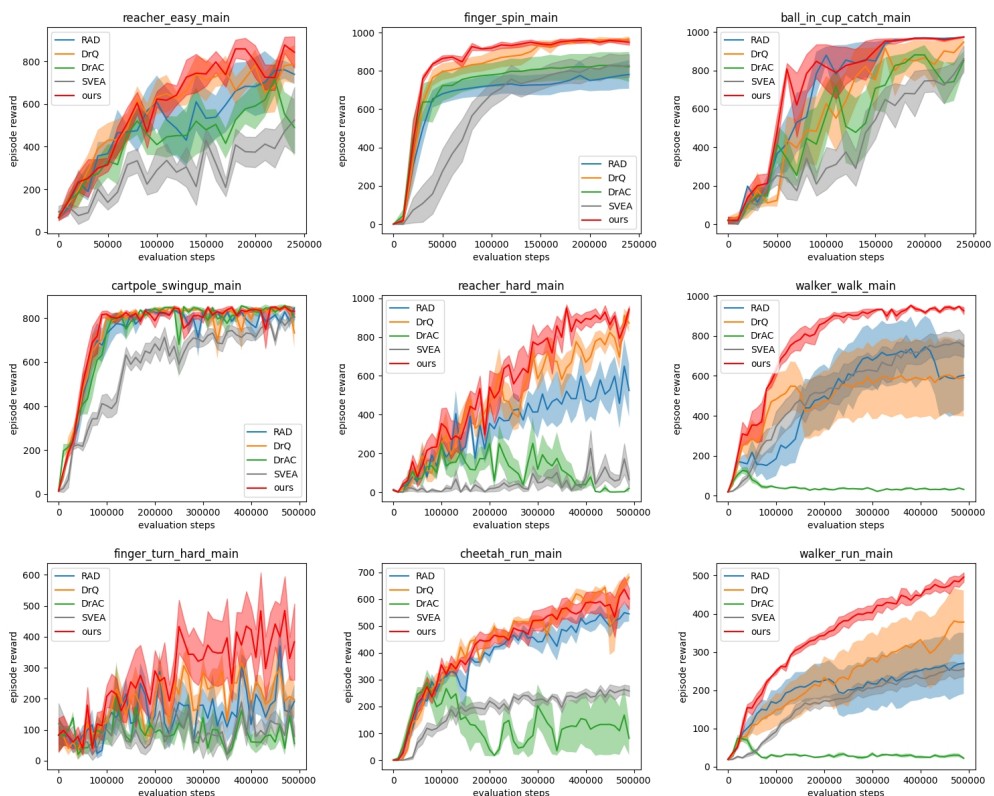

Figure 9: Comparison between different methods in DMControl with normal background.

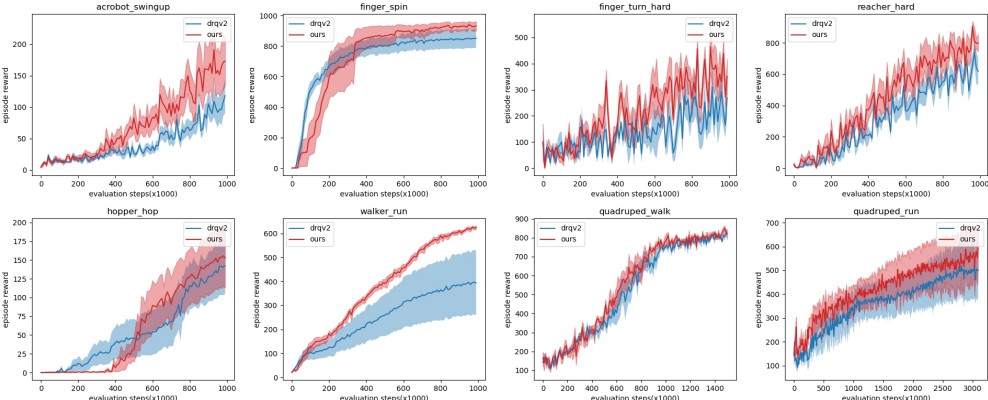

Figure 10: Results of running experiments with DDPG as base algorithm.

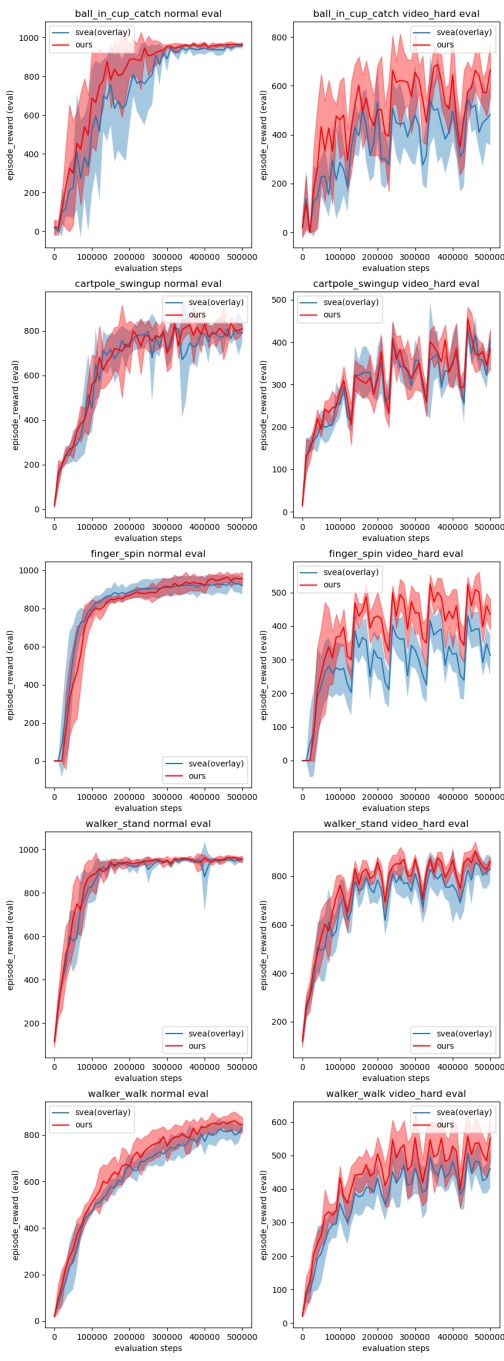

Figure 11: Comparison between SVEA and our method in DMControl with normal and video-hard backgrounds. Both methods use random overlay as image transformation. We can see the improvement in generalization ability especially in environments such as ball in cup catch, finger spin and walker walk. Since the evaluation curves are not stable even at the end of training, the recorded score in Table 2 is the average over the last 15 evaluation scores.

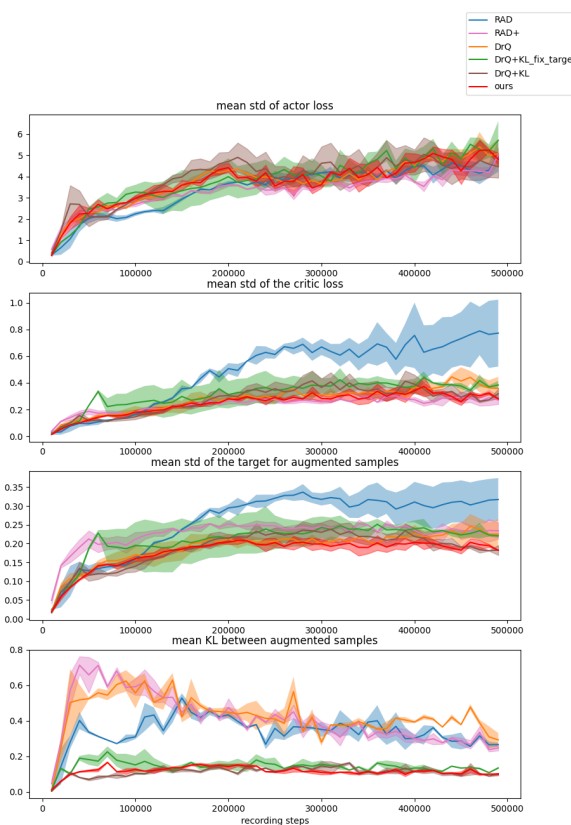

Figure 12: Some important statistics recorded along the training. The variance of critic loss and target values decreased after using more augmented samples in the training of the critic. Adding the KL divergence term to the loss can quickly enforce the invariance of the actor even at the beginning of the training.

which may lead to lower/bad performance if they are not satisfied in the real application domain. For instance, random shift/crop, which has been shown to be very effective in DMControl tasks, may yield worse performance if the agent is not well-centered in the image, according to the empirical results from Tomar et al. (2022). A better understanding of why a data augmentation transformation works in DRL is needed.

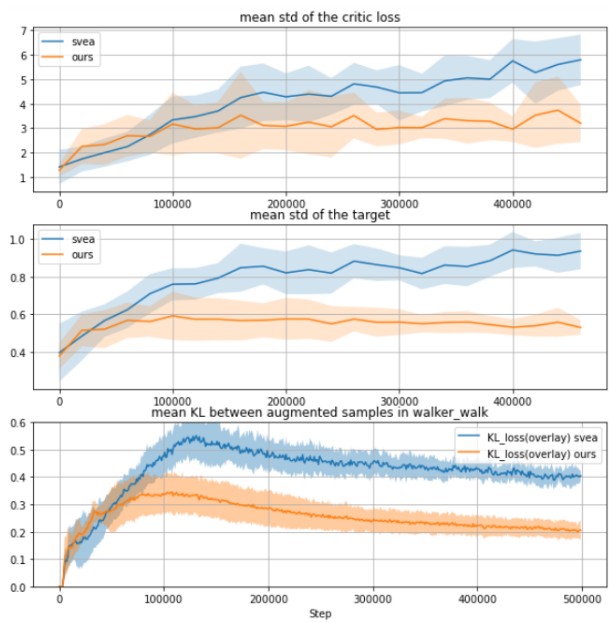

Figure 13: Some important statistics recorded along the training of SVEA and our method. With the help of KL loss and tangent prop loss, the variance of critic loss and target values are lower. Applying KL loss can quickly enforce the invariance of the actor.

