# OpenReview forum: "Revisiting Data Augmentation in Deep Reinforcement Learning"
_ICLR.cc/2024/Conference — ICLR 2024 poster_

### Official Review · Reviewer_pcif · 2023-10-28

**Soundness:** 3 good
**Presentation:** 2 fair
**Contribution:** 2 fair
**Rating:** 6
**Confidence:** 4

**Summary:**

This paper presents an analysis of various data augmentation techniques applied in image-based deep reinforcement learning.
The authors classify these techniques into two primary categories: implicit and explicit data augmentation where implicit DA consists of directly applying transformations on the observations during training, oppositely to explicit DA where transformations are applied on samples used for an auxiliary regularization term.
Following this classification, the authors then put forth a generalized formulation for integrating DA within actor-critic methods, covering both the previously mentioned implicit and explicit DA techniques. One of the paper's most salient contributions lies in the analysis of how DA impacts the variance associated with the actor and critic losses.
The insights drawn from this analysis guide the authors towards designing a principled data augmentation strategy.
A notable aspect of their proposed method is the integration of a computer vision regularization called tangent prop into the learning objective for the critic. This modification is suggested to potentially improve the stability and efficiency of the learning process.
The experimental segment of the paper includes tests on the DeepMind control suite benchmark to evaluate the proposed data augmentation scheme. Additionally, the authors test the generalization capabilities of their method using the dmcontrol-generalization-benchmark.

**Strengths:**

**Unified Framework:** A significant strength of this paper is its development of a unified framework that facilitates the direct application of data augmentation in RL. This holistic approach serves to streamline various data augmentation techniques within the context of reinforcement learning.

 **Analysis on KL regularization direction:** The paper offers a commendable examination regarding the optimal direction for applying the KL regularization term. While the theoretical insights are appreciated, an empirical evaluation would have added further depth to this assessment.

 **Variance Analysis in SAC:** The comprehensive analysis of the variance associated with the different terms used in SAC when DA is applied is insightful and sparks intriguing questions that could guide future research in the domain.

**Tangent Prop regularization:** Although the tangent prop regularization has its roots in the computer vision world and might not be a groundbreaking addition to RL, its introduction in this context is promising. The authors successfully empirically demonstrate its potential benefits.

**Weaknesses:**

**Oversimplified Unified Framework:** While the proposed unified framework is a step forward, it seems overly simplified. Specifically, the representation of implicit data augmentation as solely reliant on KL regularization seems limiting. Other methods like SODA [1] and SGQN [2] incorporate data augmentation using auxiliary objectives, drawing parallels with techniques in self-supervised learning. Such nuances should have been addressed for a more comprehensive overview.

**Excessive Reference to Appendix:** The frequent deference to the appendix, especially in sections 5.2 and 6, detracts from the paper's flow. Readers are forced into continuous back-and-forth toggling, making the narrative harder to follow.

**Unverified Claims:** The assertion that introducing image transformations in the target doesn't lead to significant variance seems unsupported, particularly when considering table 5 in the appendix. Such claims should ideally be substantiated with empirical evidence.

**Comparison with RAD and SVEA:** In the original paper, SVEA, which does not employ data augmentation on the target, significantly outperforms RAD. A comment or analysis from the authors on this discrepancy would have been informative.

**Unclear Table Presentation:** Table 4 in the appendix is a bit perplexing. The logic behind measuring variance in relation to augmentations not employed during SVEA training (such as overlay, randomconv, rotation, or blur) is unclear. Additionally, the specific "DA" used here is not explicitly mentioned, leading to further confusion.

**Novelty and Contribution:** A major point of contention is the actual novelty and impact of the paper's contributions. Its principal contribution seems to revolve around the unified framework and the introduction of tangent prop in RL. Yet, the proposed principled algorithm appears to be a minor extension of DrQ by merely incorporating a KL term. Based on figures 1 and 2, the tangible benefits seem largely attributed to the tangent prop, emphasizing its crucial role. While integrating this regularization in RL is commendable, its novelty in the broader context appears somewhat limited.

**Minor comments**
 Bolding in Tables: The rationale behind the use of bolding in tables (specifically tables 4 and 5) needs clarification.
Limitations are located in the appendix these and should be explicitly referenced in the main body.
Missing legend in figure 7, what is the green line?

**Questions:**

* Based on the data from Table 2, RAD+ exhibits lower variance on the critic loss compared to DrQ. Yet, it underperforms DrQ in terms of overall results. Given that both methods utilize the SAC algorithm, this seems to counter the claim made by the authors that reducing critic loss variance leads to more stabilized training. Can the authors provide insights or explanations for this observation?

* My understanding is that the only discernible difference between the authors' method and DrQ+KL is the introduction of the tangent prop regularization. Drawing from Table 1, it appears that this regularization is the primary factor that reduces the critic loss variance, rather than the authors' generic algorithm. Could the authors comment on the role and impact of tangent prop in this context?

* The authors assert that employing random convolution in SVEA doesn't induce significant variance. However, the empirical results from Table 5 seem to suggest otherwise. Could the authors clarify?

* What drives the intuition that the KL term assists in reducing the variance of the target critic?

For a more comprehensive analysis, would it be feasible for the authors to include SVEA in the experiments of the initial experimental setup?

---

> ### Author Response · Authors · 2023-11-15
> **Official reply to Reviewer pcif [1]**
>
> Thank you for noting the generality of our proposition and providing constructive suggestions.
> Below, we provide our answers to the weaknesses and questions.
> 1. **W1:** *Oversimplified Unified Framework*
>
> Please check point 3 in our Meta Reply.
>
> 2. **W2:** *Excessive Reference to Appendix.*
>
> We thank the reviewer to have read our paper so carefully.
> We admit that it is distracting to frequently check the appendix and we apologize for this.
> As the reviewer may understand, due to the page limit, all the information cannot be included in the main text.
> We tried to maintain the coherence of our presentation as much as possible. The references to the appendix mainly provides the theoretical proofs, further empirical evidence, additional experiments, and other minor details.
> We believe that they can be skipped in a first reading.
>
> 3. **W3:** *Unverified Claims.*
>
> Note that the expectation of the target is actually very large compared to the variance of the target.
> At the end of the training (steps 500K), the mean of the target is more than 100 while the variance of the target is just about 1 for both cases of using complex image transformations and not.
> From Table 5, although we can observe a small increase in the variance of the target due to  using complex image transformation, it is still reasonable to state that this variance increase is not the dominant issue for the performance drop.
>
> We thank the reviewer for raising this point.
> We updated Table 5 to include the mean of the target.
>
> 4. **W4:** *Comparison with RAD and SVEA*
>
> Before we compare RAD and SVEA, we need to clarify that SVEA actually uses random shift in calculating the targets and only avoids those complex image transformations in calculating the targets.
> Although it is removed from their public repository in the github commit (https://github.com/nicklashansen/dmcontrol-generalization-benchmark/commit/35a1eed87f72d3c9881c425fd83604c756849e56), we checked with the authors by email to confirm that the results in the paper were generated by applying random shift in calculating the targets.
>
> Now, if we compare the generalization ability evaluated in color-hard background, we should observe that RAD (random shift) $<$ RAD (random conv) $<$ SVEA (random conv).
> The first inequality is decided by using random conv as the image transformation.
> The second inequality is caused by avoiding using random conv in calculating the target.
>
> 5. **W5:** *Unclear Table Presentation*
>
> Thank you for raising this point.
> We apologize for the confusion and clarified the formatting of Table 4.
> In this table, each column corresponds to SVEA trained with one specified image transformation (DA) and each row corresponds to the cosine similarity recorded at 100k steps between the features of augmented states.
> Here, the cosine similarities between latent features of randomly shifted images are used as the baseline for comparisons.
> By comparing elements in one column, we can observe a smaller cosine similarity between the augmented features when training SVEA with complex image transformation which indicates the invariance with respect to complex image transformation is harder to learn.
>
> 6. **W6:** *Novelty and Contribution*
>
> Please check point 1 in our Meta Reply.
>
> 7. **W7:** *Minor comments Bolding in Tables.*
>
> Thank you for your constructive suggestions.
> We initially used bold to highlight the values that are commented in the text.
> We understand this formatting may lead to some confusion.
> We have modified our paper according to the suggestions:
> - For a clear presentation, we removed bolding in Tables 1, 4, and 5 and added more explanations in the captions.
> - We included more discussions in the Limitation section and explicitly referred to it in the Conclusion section in the main text.
> - We added the missing legends for Figure 8.

---

> > ### Author Response · Authors · 2023-11-15
> > **Official reply to Reviewer pcif [2]**
> >
> > 8. **Q1:** *Based on the data from Table 2, RAD+ exhibits lower variance on the critic loss compared to DrQ. Yet, it underperforms DrQ in terms of overall results.*
> >
> >     In Figure 2, RAD+ is better than DrQ at the beginning and becomes worse later on.
> >     In general, if everything else is kept the same, a lower variance of the critic loss can lead to a more stable training and a better performance.
> >     However, the performance of an algorithm is decided by many factors.
> >     For instance, for hard tasks such as walker\_run, we observe that RAD and DrQ are not so stable.
> >     The instability of this environment might be the key cause for this result.
> >
> > 9. **Q2:** *Could the authors comment on the role and impact of tangent prop in this context?*
> >
> >     Tangent prop regularization is derived by enforcing the derivative of the critic with respect to the image transformation parameters to be zero over the whole set of parameters.
> >     From this derivation, we can easily see that the tangent prop regularization term explicitly enforces the invariance in the critic and it is important in reducing the variance of the critic.
> >     However, we can not omit the variance reduction from using more augmented samples, which is also part of the generic algorithm.
> >
> > 10. **Q3:** *The authors assert that employing random convolution in SVEA doesn't induce significant variance. However, the empirical results from Table 5 seem to suggest otherwise. Could the authors clarify?*
> >
> >     We answered this question in point 3 (answer to W3) above.
> >
> > 11. **Q4:** *What drives the intuition that the KL term assists in reducing the variance of the target critic?*
> >
> >     The KL term explicitly enforces the invariance in the actor.
> >     When calculating the target Q-values for an augmented next state, we need to sample from the current policy to choose the next action for this augmented next state.
> >     If invariance with respect to the image transformations approximately holds, the next action will be sampled from a similar distribution over actions, which leads to a smaller variance in the target values.
> >     Under the extreme case where the invariance in the actor has been satisfied, the next action will be sampled from exactly the same distribution.
> >
> > 11. **Q5:** *For a more comprehensive analysis, would it be feasible for the authors to include SVEA in the experiments of the initial experimental setup?*
> >
> >     The new results for evaluating SVEA in the initial experimental setup are updated in Figure 3 and Figure 9 in the newest version.
> >     It is not surprising that using SVEA with random overlay increases the generalization ability of the agent at the cost of sample efficiency.
> >     Especially in environments like reacher, the target goal might not be clear in the image when random overlay is applied.
> >     We also updated our paper to include the new results.
> >
> > Please let us know if you have any other comments or concerns!

---

> ### Comment · Reviewer_pcif · 2023-11-22
>
> I acknowledge the author's responses to my questions. Given this information and the reviews provided by my colleagues, I am willing to raise my score from 5 to 6.

---

> > ### Author Response · Authors · 2023-11-22
> > **Thanks for your reply!**
> >
> > Dear Reviewer pcif,
> >
> > We appreciate your response.  Thank you again for your constructive suggestions for revising our paper.
> >
> > Best,
> > Authors

---

### Official Review · Reviewer_9PSY · 2023-10-29

**Soundness:** 4 excellent
**Presentation:** 3 good
**Contribution:** 3 good
**Rating:** 6
**Confidence:** 3

**Summary:**

This paper delves into the realm of data augmentation methods in Deep Reinforcement Learning (DRL), conducting a thorough analysis of existing techniques. The authors provide a theoretical framework to understand and compare these methods and propose a new regularization term, tangent prop, to enhance invariance. Their contributions lie in offering a theoretical understanding of data augmentation methods in DRL, suggesting how to use data augmentation in a more theoretically-driven manner, and introducing a novel regularization approach. Through extensive experiments, the authors validate their theoretical propositions.

**Strengths:**

Originality:
It conducts a comprehensive analysis of existing data augmentation methods, initially based on the author's introduction of assumptions regarding uncertainty in reinforcement learning and existing image-based online Deep Reinforcement Learning (DRL) data augmentation techniques. The paper establishes an integrated AC framework incorporating data augmentation. Within this framework, the mainstream data augmentation methods are analyzed and categorized into explicit and implicit regularization techniques. Qualitative and quantitative analyses of these augmentation methods are performed. Building upon these two types of regularization, the paper introduces a novel regularization technique – tangent prop – providing theoretical support for addressing uncertainty in the Critic component.

Quality and Clarity:
I think the quality of the paper is very high. Each proposition and hypothesis presented in the paper is accompanied by corresponding formulas, and the appendix contains detailed derivations of these formulas. Additionally, I find the paper to be very logically structured.
First, the paper explains the process of the Data-Augmented Off-policy Actor-Critic Scheme and introduces the key hypotheses regarding the uncertainty in Q-values and policy π. The subsequent proofs and derivations are based on these definitions. The paper then elaborates on how explicit and implicit uncertainties are defined within the A-C framework and how they affect the Loss function of Actor and Critic. After providing the theoretical background, the paper proposes its own generic algorithm and provides detailed derivations and proofs.

Furthermore, the paper analyzes the effects of applying different image transformations when calculating target Q-values, as well as the empirical actor/critic losses estimated under data augmentation. In implicit regularization, the author, unlike the previous SVEA method, incorporates KL divergence into the training process for policy. The paper explains and proves that, under the premise of Critic invariance, introducing KL divergence helps the model better learn the invariance of the Actor. Finally, the paper introduces the innovative concept of Tangent Propagation to further demonstrate how this newly introduced additional regularization term promotes Critic invariance.

The logic throughout the main body of the paper is very clear. It systematically proves its hypotheses about uncertainty and effectively addresses the initial problems posed in the paper.

Significance:
The paper holds significant value for the DRL research community. By offering a theoretical framework and introducing a novel regularization approach, the paper addresses a key challenge in DRL, enhancing the understanding and application of data augmentation techniques. The experimental validation across various environments supports the significance of the proposed methods. The findings are likely to influence future research directions, providing valuable insights for researchers and practitioners in the field.

**Weaknesses:**

The theoretical derivation in the paper is very thorough. However, I believe the experimental section of the paper is somewhat lacking. It compares the performance with the previous statistically trained model, SVEA, providing detailed experimental data and theoretical analysis. Nevertheless, there is a lack of in-depth analysis of the shortcomings of the previous algorithms and the advantages of the proposed algorithm. Moreover, I think the algorithm should be further compared with more methods and applied in various domains to validate its generalizability.

The entire article is dedicated to theoretically proving the reliability of its algorithm, and it has obtained favorable experimental results. However, throughout the entire text, there is no discussion about the shortcomings and limitations of the algorithm, nor is there any mention of how to extend the algorithm or areas that need further exploration in the future.

The article discusses how different complex image augmentation techniques have varying impacts on the invariance of target Q-values. The paper proposes using cosine similarity measured from encoder outputs for this evaluation. It is mentioned in the article that techniques such as random addition or Gaussian blur easily achieve invariance through these transformations because they result in high cosine similarity. When using these types of image transformations in computing target values, it does not affect performance. In contrast, image transformations like random convolution or random rotation are relatively challenging to achieve invariance during the training process, as they result in lower cosine similarity. The solution proposed in the paper is to enforce this invariance by performing more updates at each training step. However, the paper does not address whether there might be potential overfitting issues under these specific conditions and how to handle them.

**Questions:**

Refer to the weaknesses.

---

> ### Author Response · Authors · 2023-11-15
> **Official reply to Reviewer 9PSY**
>
> Thank you for noting the quality of our work especially in originality and significance.
> We are happy that you recognize our paper being logically clear.
> Below, we provide an answer to the weaknesses.
>
> 1. **W1:** *However, I believe the experimental section of the paper is somewhat lacking.*
>
> To the best of our knowledge, most recently-proposed techniques for improving generalization ability are actually orthogonal to our analysis and could be combined with our  proposed algorithm.
> However, if we have missed any  relevant work, we would appreciate if you could provide some specific references and we would try our best to add some additional comparative experimental results in our final version.
> As for the limitations of our proposed algorithm, please check point 2 in our Meta Reply.
>
> 2. **W2:** *However, throughout the entire text, there is no discussion about the shortcomings and limitations of the algorithm, nor is there any mention of how to extend the algorithm or areas that need further exploration in the future.*
>
> Please check point 2 in our Meta Reply.
>
> 3. **W3:** *However, the paper does not address whether there might be potential overfitting issues under these specific conditions and how to handle them.*
>
> We did not observe this overfitting issue when we updated more.
> To verify this point, we ran some additional experiments to provide further evidence for it.
> First, we can see that the training scores are similar to the evaluation scores when we update more, as shown in the figure (evaluation_training_score_for_reviewer_9PSY.png) in our new supplementary material.
> We also ran the experiments of training the agent with more iterations when more updates are applied per iteration and the result is shown in the figure (More_iterations_and_more_updates_for_reviewer_9PSY.jpeg) in our new supplementary material.
> In this experiment, we trained the agent with more iterations and more updates per iterations while we did not observe an overfitting issue.
>
> Please let us know if you have any other comments or concerns!

---

> > ### Author Response · Authors · 2023-11-22
> > **Kind reminder**
> >
> > Dear reviewer 9PSY,
> >
> > We believe we have addressed all your concerns. We have improved our submission thanks to your feedback. Please let us know if you have any other concerns, which we would be happy to address.
> >
> > Best, Authors

---

### Official Review · Reviewer_BeKW · 2023-10-30

**Soundness:** 3 good
**Presentation:** 2 fair
**Contribution:** 3 good
**Rating:** 6
**Confidence:** 5

**Summary:**

This paper theoretically analyzes and compares the impact of current data augmentation techniques on Q-target and actor/critic loss in visual reinforcement learning. The regularization term "tangent prop" proposed in computer vision is also applied to reinforcement learning to learn invariant.

**Strengths:**

The theoretical analysis of the paper is sufficient, which is difficult to see in many similar works. At the same time, the experimental data are also considerable.

**Weaknesses:**

1) The work in this paper seems to be equivalent to adding an explicit regularization to implicit regularization, is it equivalent to DrQ combined with DrAC in terms of functionality?
2) How the proposed algorithm addresses the initial question "Although they empirically demonstrate the effectiveness of data augmentation for improving sample efficiency or generalization, which technique should be preferred is not always clear.". The main idea of the paper is not analyzed.

**Questions:**

1) Please check that the two terms in the KL divergence in Eq. 8, and there seems to be an ambiguity with Eq. 4 and Eq. 9.
2) How to prove that the distributions of $\nu$ and $\mu$ in Eq. 11 and Eq. 12 are the distributions of $\hat{\nu}$ and $\hat{\mu}$ satisfying Lemma 1?
3) How to show that "even using complex image transformations such as random convolution in the target". We did not find an experiment in the paper that verifies this conclusion.
4) In the experimental part, the grid search for $ \alpha_{KL}$ and $ \alpha_{tp}$ is too sparse, and the final choice of 0.1 leads to curiosity about the results within [0,0.1]. A similar situation occurs with the SVEA comparison experiment of 0.5 selected from {0.1, 0.5}.
5) I'm curious about the method of determining $ \alpha_i$, which seems to be missing the reason(s) in the paper. Does it have a specific value for each augmentation in the set, or does it choose an average weight?
6) In Figure 2, the batch size of DrQ reproduced in the results is 256 instead of 512 in the original DrQ. The scores of DrQ will decrease under some environments when using a smaller batch size. Compared with the official results of DrQ (https://github.com/denisyarats/drq), some results of DrQ shown in Figure 9 are lower. Such as ball_in_catch , walker_walk, walk_run. Therefore, to make a fair comparison, the author should completely use the hyperparameter settings in DrQ to reproduce, or use the scores in DrQ.

---

> ### Author Response · Authors · 2023-11-15
> **Official reply to Reviewer BeKW [1]**
>
> Thank you for summarizing our work and highlighting that our theoretical analysis is rare and valuable compared to many similar work.
> Below, we provide our answer to the weaknesses and questions.
>
> 1. **W1:** *The work in this paper seems to be equivalent to adding an explicit regularization to implicit regularization, is it equivalent to DrQ combined with DrAC in terms of functionality?*
>
> Please check point 1 in our Meta Reply.
>
> Roughly speaking, our method follows the idea from DrQ to apply invariant image transformations in both the training of the actor and critic, complements it with a KL regularization term in the actor loss and introduces a new regularization term called tangent prop in the critic loss.
> However, there are three key differences between our method and the simple combination of DrQ and DrAC:
> - Our method introduces a new regularization term called tangent prop which is novel in DRL.
> - The KL regularization term in our method is not exactly the KL regularization term used in DrAC.
> The KL regularization term $D_{KL}[\pi_\theta(\cdot|s),\pi_\theta(\cdot|f_\mu(s))]$ used in DrAC only augments the state on one side while the KL regularization term $D_{KL}[\pi_\theta(\cdot|f_\eta(s),\pi_\theta(\cdot|f_\mu(s))]$ in our method augments the states on both sides.
> The regularization term in our method is somehow equivalent to using an averaged policy as the target, which may be preferred.
> The detailed analysis is presented at the end of Section 4.2 and the experimental comparisons between these two KL regularizations are shown in Figure 2. The two lines which are respectively labeled as KL\_fix\_target and KL correspond to the KL from DrAC and the KL in our method.
> - Our method is compatible with using more than one type of image transformation in the training which is required to achieve state-of-the-art performance in both sample efficiency and generalization ability.
> Moreover, we also provide an analysis and suggest a criterion for the usage of complex image transformations.
>
> 2. **W2:** *How the proposed algorithm addresses the initial question "Although they empirically demonstrate the effectiveness of data augmentation for improving sample efficiency or generalization, which technique should be preferred is not always clear.".*
>
> The goal of this work was to analyze existing methods and make recommendations on how to use data augmentations in a more principled way in DRL.
> We start our analysis by connecting the implicit with explicit regularization and indicate some preferences about them.
> For example, the actor loss in implicit regularization can only enforce the invariance in the actor under the condition that the invariance in the critic has been learned.
> This motivates our inclusion of an explicit KL regularization term in the final actor loss of our method.
> Note that we justify most of our recommendations in Section 5.2.
> For example, we recommend applying complex image transformations in calculating the target Q values and provide a criterion for judging if an image transformation should be considered as complex or not.
> All in all, we tried to judge different techniques with our theoretical analysis and validate our propositions empirically.
>
> 3. **Q1:** *Please check that the two terms in the KL divergence in Eq. 8, and there seems to be an ambiguity with Eq. 4 and Eq. 9.*
>
> There is actually no ambiguity here.
> The KL term in Equation 8 is different from the KL terms in Equation 4 and Equation 9, as listed below:
> - Only the state in one side of the KL term in Equation 8 is augmented while the states in both sides of the KL term in Equation 4/Equation 9 are augmented.
> The differences between them are explained in our answer 1(b) to W1 above.
> - The directions of the KL regularization terms are different.
> In Equation 8, the stop gradient is attached to the $\theta$ on the right hand side.
> Considering that KL is asymmetric, it is necessary to discuss the difference which is presented in the second paragraph below Proposition 4.1.
> In conclusion, we found that using the stop gradient in the first term is better and including an explicit KL term in the actor loss is preferred.
>
> 4. **Q2:** *How to prove that the distributions of $\nu$ and $\mu$ in Eq. 11 and Eq. 12 are the distributions of $\hat\nu$ and $\hat\mu$ satisfying Lemma 1?*
>
>     Note that in Eqns. 11 and 12, the distributions of $\nu$ and $\mu$ are not restricted. They are usually set to be uniform distributions, but other distributions could be chosen depending on the application domain.
>
>     Lemma 1 simply states that for any distributions $\nu$, implicit regularization in the critic with specifically chosen $\hat\nu$ and $\hat\mu$ (see Proof of Lemma 1 for details) is equivalent to explicit regularization in the critic with $\nu$.

---

> ### Author Response · Authors · 2023-11-15
> **Official reply to Reviewer BeKW [2]**
>
> 5. **Q3:** *How to show that "even using complex image transformations such as random convolution in the target". We did not find an experiment in the paper that verifies this conclusion.*
>
> We apologize for the ambiguity here. The experimental results for this point are shown in Appendix F.
> We modified our paper to explicitly indicate that all observations about using complex image transformations in calculating the targets are presented in Appendix F.
> Moreover, we clarified the descriptions of Tables 4 and 5.
>
> 6. **Q4:** *In the experimental part, the grid search for $\alpha_{KL}$ and $\alpha_{tp}$
>      is too sparse, and the final choice of 0.1 leads to curiosity about the results within [0,0.1]. A similar situation occurs with the SVEA comparison experiment of 0.5 selected from {0.1, 0.5}.*
>
> Considering that our main goal was to provide some justifications and recommendations on techniques used in data augmentation methods in DRL, we did not focus on getting the best possible score on this DMControl benchmark.
> However, we can indeed perform a more thorough grid search.
> Due to the time constraint in this rebuttal phase, we will update the results in the final version.
>
> 7. **Q5:** *I'm curious about the method of determining $\alpha_i$, which seems to be missing the reason(s) in the paper. Does it have a specific value for each augmentation in the set, or does it choose an average weight?*
>
> We simply use an average weight for different image transformations, as it was done in SVEA.
>
> 8. **Q6:** *In Figure 2, the batch size of DrQ reproduced in the results is 256 instead
>     of 512 in the original DrQ.*
>
> We ran the experiments of DrQ and our method in the environments listed in the question with batch size of 512:
> |  Envs  | DrQ(batch_size=256) | DrQ(batch_size=512) | Ours(batch_size=256) | Ours(batch_size=512)
> |-----------------|-------------------|----------------|----------------|----------------|
> | Ball in cup catch | 947.38+-41.76  |962.08+-19.54 |974.06+-5.09|969.96+-8.30
> | Walker walk | 591.25+-433.38 | 740.67+-343.23 |925.62+-35.03 |903.11+-56.54
> | Walker run | 378.29+-184.58 | 350.71+-210.66 |496.06+-26.33 |569.17+-63.75
>
> We can see that
> 1. Our method is insensitive to the choice of batch size.
> 2. Without retuning the hyperparameter, our method always matches or outperforms DrQ in these environments with these two choices of batch size.
> 3. Considering the modest computational resources we have and the huge increase in computational cost due to usage of batch size = 512, it is fair to compare sample efficiency of all the methods with the batch size of 256.
>
> Please let us know if you have any other comments or concerns!

---

> > ### Author Response · Authors · 2023-11-22
> > **Kind reminder**
> >
> > Dear reviewer BeKW,
> >
> > We believe we have addressed all your concerns. We have improved our submission thanks to your feedback. Please let us know if you have any other concerns, which we would be happy to address.
> >
> > Best, Authors

---

> > > ### Comment · Reviewer_BeKW · 2023-11-23
> > >
> > > Thanks for the author's responses to my comments. Based on the comments of all reviewers and the author's reply, I decided to keep my score.

---

> > > > ### Author Response · Authors · 2023-11-23
> > > > **Thanks for your reply**
> > > >
> > > > Dear reviewer BeKW,
> > > >
> > > > We appreciate your help in improving our paper. Thank you again for your reply.
> > > >
> > > > Best,
> > > > Authors

---

### Official Review · Reviewer_GnUL · 2023-11-08

**Soundness:** 3 good
**Presentation:** 3 good
**Contribution:** 3 good
**Rating:** 6
**Confidence:** 3

**Summary:**

The paper revisits state-of-the-art data augmentation methods in Deep Reinforcement Learning (DRL). It offers a theoretical analysis to understand and compare different existing methods, providing recommendations on how to exploit data augmentation in a more theoretically-motivated way. The paper introduces a novel regularization term called tangent prop and validates the propositions by evaluating the method in DeepMind control tasks. The experimental results are to be released after publication, and the paper also includes theoretical proofs to support the proposed methods and analysis. However, it is important to note that the paper is highly technical and may be challenging for readers without a strong background in DRL and computer vision to understand. Additionally, it focuses on image-based DRL and does not address the ethical implications of using data augmentation techniques in DRL.

**Strengths:**

1.  The paper provides a comprehensive analysis of existing data augmentation techniques in DRL and offers recommendations on how to use them more effectively.
2.  The authors introduce a novel regularization term called tangent prop and demonstrate its state-of-the-art performance in various environments.
3.  The experimental results are presented in a clear and concise manner, and the code with comments on how to reproduce the results will be released after publication.
4.  The paper provides theoretical proofs to support the proposed methods and analysis.

**Weaknesses:**

1. More insights into the limitations and potential failures of the proposed method should be discussed. This would provide a more balanced perspective and help readers better understand the practical considerations when applying the proposed approach.

2. Further analysis and comparisons with a wider range of existing techniques should be conducted to showcase the advantages and limitations of the proposed method in different scenarios. This would provide a more comprehensive view of its effectiveness and contribution to the field.

**Questions:**

NA

---

> ### Author Response · Authors · 2023-11-15
> **Official reply to Reviewer GnUL**
>
> Thank you for indicating our contributions of comprehensive theoretical analysis and concise experimental validations.
> Below, we provide an answer to the points corresponding to the weaknesses.
>
>
> 1. **W1:** *More insights into the limitations and potential failures of the proposed method should be discussed.*
>
> Please check point 2 in our Meta Reply.
>
> 2. **W2:** *Further analysis and comparisons with a wider range of existing techniques should be conducted to showcase the advantages and limitations of the proposed method in different scenarios.*
>
> Please check point 3 in our Meta Reply.
>
> Please let us know if you have any other comments or concerns!

---

> > ### Author Response · Authors · 2023-11-22
> > **Kind reminder**
> >
> > Dear reviewer GnUL,
> > We believe we have addressed all your concerns. We have improved our submission thanks to your feedback. Please let us know if you have any other concerns, which we would be happy to address.
> >
> > Best,
> > Authors

---

### Author Response · Authors · 2023-11-15
**Meta Reply**

We would like to thank all the reviewers for their effort in reviewing our paper and providing us helpful feedback.
Following their suggestions, we improved our paper accordingly (see latest submission).
Below, we provide an answer to three common points raised about novelty, limitations, and related work with self-supervised losses.

**Novelty:**

We want to emphasize again that our goal is to analyze existing methods and provide theoretically-sound recommendations on how to apply data augmentations in DRL.
Both our theoretical analysis and empirical evaluation suggest that our proposed algorithm is superior to previously-proposed methods that enforce invariance in the actor/critic networks.
In addition to our proposed algorithm, we believe that our theoretical analysis (which helps gain a better understanding of data augmentation in DRL) and our discussion about complex transformations (which actually contradicts previous work) are also valuable.

**Limitations:**

Due to the page limit, we could unfortunately not provide a comprehensive analysis for the limitations in the main text.
Here, we briefly list several limitations and potential failures of the proposed method, which we will include in the Limitations section in the appendix.
1. Our method naturally requires the knowledge of some  effective image transformations for a given task.
Without such knowledge, the invariant transformations for a  problem would need to be learned, which is currently an active research direction.
2. Image transformation may rely on some implicit assumptions, which may lead to lower/bad performance if they are not satisfied in the real application domain.
For instance, random shift/crop, which has been shown to be very effective in DMControl tasks, may yield worse performance if the agent is not well-centered in the image, according to the empirical results from [1].
A better understanding of why a data augmentation transformation works in DRL is needed.

**Methods with self-supervised losses:**

Our goal was to analyze the methods that apply data augmentation in image-based DRL to directly enforce invariance in the outputs of the actor and critic networks.
The work related to using self-supervised losses in DRL is not included in this paper for several reasons:
1. In contrast to our focus, the self-supervised losses are usually applied on the latent space to enforce invariance or add constraints on the latent features.
2. Data augmentation usually serves as one technique among others in these methods.
For example, in SPR [2] the key contribution is to incorporate temporal information in the training of the encoder, while in SGQN [3] it is to learn a saliency map to guide the agent to focus on important information in the image.
In both these methods, image transformations are used for creating different views of the states/images.
Note that using a self-supervised loss alone may be insufficient [4].

Therefore, it is not clear how the methods using self-supervised losses in DRL and those we discussed in our paper can be both studied in a unified framework, since they are different in nature.
To the best of our knowledge, we believe that we have included all the most related work.
However, if you have any further references for other relevant work that fits our framework, we would be happy to include their analysis in the final version of our paper.

Theoretically analyzing the methods using self-supervised losses in DRL would indeed be very interesting.
We leave it for future work.

[1] Manan Tomar, Utkarsh Aashu Mishra, Amy Zhang, and Matthew E. Taylor. Learning representations for pixel-based control: What matters and why? [TMLR 2023]

[2] Max Schwarzer, Ankesh Anand, Rishab Goel, R Devon Hjelm, Aaron Courville, and Philip Bachman. Data-Efficient Reinforcement Learning with Self-Predictive Representations. [ICLR 2021]

[3] David Bertoin, Adil Zouitine, Mehdi Zouitine, and Emmanuel Rachelson. Look where you look! Saliency-guided Q-networks for generalization in visual reinforcement learning. [NeurIPS 2022]

[4] Xiang Li, Jinghuan Shang, Srijan Das, Michael S Ryoo. Does self-supervised learning really improve reinforcement learning from pixel? [NeurIPS 2022]

---

### Meta-Review · Area_Chair_z454 · 2023-12-04

**Metareview:**

This paper contributes in the following way:
1. A theoretical framework for unifying previous RL-data augmentation techniques (ex: RAD, DrQ, SVEA, DrAC)
2. Derivation and motivation of certain regularization terms from agent invariance assumptions (KL and Tangent-Prop terms)
3. Experimental results over DM-Control demonstrating improvements over previous data augmentation methods, and ablations over appendix.

All reviewers agree that the paper is well written, provides theoretical grounding and framing of previous data augmentation techniques, and solid experiments and should be accepted.

I would recommend the authors consider emphasizing more significant and impactful aspects of the paper (the theoretical analysis and shortcomings of previous methods) in the main body.

**Justification For Why Not Higher Score:**

Beyond the theoretical justifications, it could be argued that the method-specific contributions (i.e. additional KL and Tangent Prop terms) are not incredibly groundbreaking and somewhat straightforward (i.e. clearly one should regularize the agent to be invariant to image transformations). This was a point raised by Reviewers BeKW and pcif.

**Justification For Why Not Lower Score:**

The paper provides solid new contributions (clear theoretical analysis, new proposed regularization terms, higher performance than previous baselines) which merit it at least acceptance.

---

### Decision · Program_Chairs · 2024-01-16

Accept (poster)